# How does This Interaction Affect Me? Interpretable Attribution for Feature Interactions

**Michael Tsang, Sirisha Rambhatla, Yan Liu**
Department of Computer Science
University of Southern California
{tsangm,sirishar,yanliu.cs}@usc.edu

## Abstract

Machine learning transparency calls for interpretable explanations of how inputs relate to predictions. Feature attribution is a way to analyze the impact of features on predictions. Feature *interactions* are the contextual dependence between features that jointly impact predictions. There are a number of methods that extract feature interactions in prediction models; however, the methods that assign attributions to interactions are either uninterpretable, model-specific, or non-axiomatic. We propose an interaction attribution and detection framework called `Archipelago` which addresses these problems and is also scalable in real-world settings. Our experiments on standard annotation labels indicate our approach provides significantly more interpretable explanations than comparable methods, which is important for analyzing the impact of interactions on predictions. We also provide accompanying visualizations of our approach that give new insights into deep neural networks.

## 1   Introduction

The success of state-of-the-art prediction models such as neural networks is driven by their capability to learn complex feature interactions. When such models are used to make predictions for users, we may want to know how they personalize to us. Such model behavior can be explained via *interaction detection* and *attribution*, i.e. if features influence each other and how these interactions contribute to predictions, respectively. Interaction explanations are useful for applications such as sentiment analysis [36], image classification [48], and recommendation tasks [21, 48].

Relevant methods for attributing predictions to feature interactions are black-box explanation methods based on axioms (or principles), but these methods lack interpretability. One of the core issues is that an interaction's importance is not the same as its attribution. Techniques like Shapley Taylor Interaction Index (STI) [14] and Integrated Hessians (IH) [25] combine these concepts in order to be axiomatic. Specifically, they base an interaction's attribution on non-additivity, i.e. the degree that features non-additively affect an outcome. While non-additivity can be used for interaction detection, it is not interpretable as an attribution measure as we see in Fig. 1. In addition, neither STI nor IH is tractable for higher-order feature interactions [14, 46]. Hence, there is a need for interpretable, axiomatic, and scalable methods for interaction attribution and corresponding interaction detection.

To this end, we propose a novel framework called `Archipelago`[1], which consists of an interaction attribution method, `ArchAttribute`, and a corresponding interaction detector, `ArchDetect`, to address the challenges of being interpretable, axiomatic, and scalable. `Archipelago` is named after its ability to provide explanations by isolating feature interactions, or feature "islands". The inputs to `Archipelago` are a black-box model $f$ and data instance $\mathbf{x}^{\star}$, and its outputs are a set of interactions and individual features $\{\mathcal{I}\}$ as well as an attribution score $\phi(\mathcal{I})$ for each of the feature sets $\mathcal{I}$.

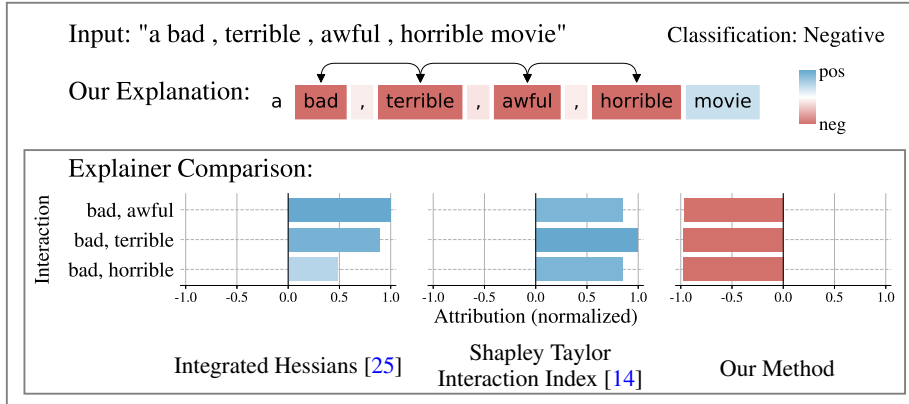

Figure 1: Our explanation for the sentiment analysis example of [25]. Colors indicate sentiment, and arrows indicate interactions. Compared to other axiomatic interaction explainers, only our work corroborates our intuition by showing negative attribution among top-ranked interactions.

`ArchAttribute` satisfies attribution axioms by making relatively mild assumptions: a) disjointness of interaction sets, which is easily obtainable, and b) the availability of a generalized additive function which is a good approximator to any function [49–51]. On the other hand, `ArchDetect` circumvents intractability issues of higher-order interaction detection by removing certain uninterpretable higher-order interactions and leveraging a property of feature interactions that allows pairwise interactions to merge for disjoint arbitrary-order interaction detection. In practice, where any assumptions may not hold in real-world settings, `Archipelago` still performs well. In particular, `Archipelago` effectively detects relevant interactions and is more interpretable than state-of-the-art methods [14, 20, 25, 26, 47, 51] when evaluated on annotation labels in sentiment analysis and image classification. We visualize `Archipelago` explanations on sentiment analysis, coronavirus detection on chest X-rays, and ad-recommendation, and we demonstrate an interactive visualization of `Archipelago`.

Our main contributions are summarized below.

- **Interaction Attribution:** We propose `ArchAttribute`, a feature attribution measure which leverages feature interactions. It has advantages of being model-agnostic, interpretable, and runtime-efficient as compared to other state-of-the-art interaction attribution methods.
- **Principled Attribution:** `ArchAttribute` obeys standard attribution axioms [47] that are generalized to work for feature sets, and we also propose a new axiom for interaction attribution to respect the additive structure of a function.
- **Interaction Detection:** We propose a complementary feature interaction detector, `ArchDetect`, which is also model-agnostic and $\mathcal{O}(p^2)$-efficient for pairwise and disjoint arbitrary-order interaction detection ($p$ is number of features).

`Archipelago` satisfies other desirable qualities of interpretability [34]: *coherence*, *simplicity*, *generality*, and *truthfulness*. We achieve *coherence* by separating interaction attribution from detection, *simplicity* by merging feature sets, *generality* by isolating attributions from contexts, and *truthfulness* by trying to explain the true cause of predictions via interactions. Our empirical studies on `Archipelago` demonstrate its superior properties as compared to state-of-the-art methods.

## 2 Preliminaries

We first introduce preliminaries that serve as a basis for our discussions.

**Notations:** We use boldface lowercase symbols, such as $\mathbf{x}$, to represent vectors. The $i$-th entry of a vector $\mathbf{x}$ is denoted by $x_i$. For a set $\mathcal{S}$, its cardinality is denoted by $|\mathcal{S}|$, and the operation $\setminus \mathcal{S}$ means all except $\mathcal{S}$. For $p$ features in a dataset, let $\mathcal{I}$ be a subset of feature indices: $\mathcal{I} \subseteq \{1, 2, \ldots, p\}$. For a vector $\mathbf{x} \in \mathbb{R}^p$, let $\mathbf{x}_{\mathcal{I}} \in \mathbb{R}^p$ be defined element-wise in (1). In our discussions, a *context* means $\mathbf{x}_{\setminus \mathcal{I}}$.

$$(\mathbf{x}_{\mathcal{I}})_i = \left\{ \begin{array}{ll} x_i, & \text{if } i \in \mathcal{I} \\ 0 & \text{otherwise} \end{array} \right. \tag{1}$$

**Problem Setup:** Let $f$ denote a black-box model with scalar output. For multi-class classification, $f$ is assumed to be a class logit. We use an input vector $\mathbf{x}^\star \in \mathbb{R}^p$ to denote the data instance where we

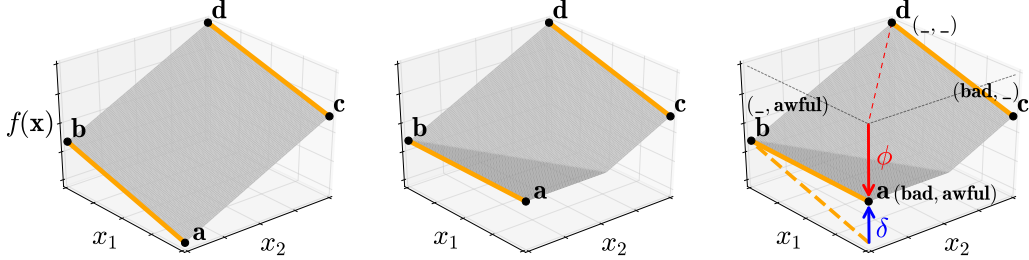

(a) Additive (linear) function    (b) Non-additive (ReLU) function  (c) $\delta$ vs. $\phi$ on a text example (Fig. 1)

Figure 2: Non-additive interaction for $p = 2$ features: The corner points are used to determine if $x_1$ and $x_2$ interact based on their non-additivity on $f$, i.e. they interact if $\delta \propto (f(\mathbf{a}) - f(\mathbf{b})) - (f(\mathbf{c}) - f(\mathbf{d})) \neq 0$ (§4.1). In (c), the attribution of (bad, awful) should be negative via $\phi$ (2), but Shapley Taylor Interaction Index uses the positive $\delta$. Note that $\phi$ depends on $\mathbf{a}$ and $\mathbf{d}$ whereas $\delta$ depends on $\mathbf{a}$, $\mathbf{b}$, $\mathbf{c}$, and $\mathbf{d}$. Also, Integrated Hessians is not relevant here since it does not apply to ReLU functions.

wish to explain $f$, and $\mathbf{x}' \in \mathbb{R}^p$ to denote a *neutral baseline*. Here, the baseline is a reference vector for $\mathbf{x}^\star$ and conveys an "absence of signal" as per [47]. These vectors form the space of $\mathcal{X} \subset \mathbb{R}^p$, where each element comes from either $x_i^\star$ or $x_i'$, i.e. $\mathcal{X} = \{(x_1, \ldots, x_p) \mid x_i \in \{x_i^\star, x_i'\}, \forall i = 1, \ldots, p\}$.

**Interpretability:** One of the motivations of this work is to provide coherent interpretability, meaning that it needs to be consistent with human prior belief [34]. Coherence is important so that users can agree with model interpretations and don't inadvertently think a model fails when it actually works, as we motivate in Fig. 1. Coherent interpretability leads to more trustworthy explanations [26]. In our experiments, we leverage standard annotation labels to measure coherence.

**Feature Interaction:** The definition of the feature interaction of interest is formalized as follows.

**Definition 1** (Statistical Non-Additive Interaction). *A function $f$ contains a statistical non-additive interaction of multiple features indexed in set $\mathcal{I}$ if and only if $f$ cannot be decomposed into a sum of $|\mathcal{I}|$ subfunctions $f_i$ , each excluding the $i$-th interaction variable: $f(\mathbf{x}) \neq \sum_{i \in \mathcal{I}} f_i(\mathbf{x}_{\setminus\{i\}})$.*

Def. 1 identifies a non-additive effect among all features $\mathcal{I}$ on the output of function $f$ [18, 46, 49]. For example, this means that the function $\text{ReLU}(x_1 + x_2)$ creates a feature interaction because it cannot be represented as an addition of univariate functions, i.e. $\text{ReLU}(x_1 + x_2) \neq f_1(x_2) + f_2(x_1)$ (Fig. 2b). We refer to individual feature effects which do not interact with other features as *main effects*. Higher-order feature interactions are captured by $|\mathcal{I}| > 2$, i.e. interactions larger than pairs. Additionally, if a higher-order interaction exists, all of its subsets also exist as interactions [46, 49].

## 3  `Archipelago` Interaction Attribution

We begin by presenting our feature attribution measure. Our feature attribution analyzes and assigns scores to detected feature interactions. Our corresponding interaction detector is presented in §4.

### 3.1  ArchAttribute

Let $\mathcal{I}$ be the set of feature indices that correspond to a desired attribution score. Our proposed attribution measure, called `ArchAttribute`, is given by

$$\phi(\mathcal{I}) = f(\mathbf{x}_{\mathcal{I}}^\star + \mathbf{x}_{\setminus\mathcal{I}}') - f(\mathbf{x}'). \tag{2}$$

`ArchAttribute` essentially isolates the attribution of $\mathbf{x}_{\mathcal{I}}^\star$ from the surrounding baseline context $\mathbf{x}_{\setminus\mathcal{I}}'$ while also satisfying axioms (§3.2). We call this isolation an "island effect", where the input features $\{x_i^\star\}_{i \in \mathcal{I}}$ do not specifically interact with the baseline features $\{x_j'\}_{j \in \setminus\mathcal{I}}$. For example, consider sentiment analysis on a phrase $\mathbf{x}^\star = $ "not very bad" with a baseline $\mathbf{x}' = $ "_ _ _" . Suppose that we want to examine the attribution of an interaction $\mathcal{I}$ that corresponds to {very, bad} in isolation. Here, the contextual word "not" also interacts with $\mathcal{I}$, which becomes apparent when small perturbations to the word "not" causes large changes to prediction probabilities. However, as we move further away from the word "not" towards the empty-word "_" in the word-embedding space, small perturbations no longer result in large prediction changes, meaning that the "_" context does not specifically

interact with {very, bad}. This intuition motivates our use of the baseline context $\mathbf{x}'_{\setminus \mathcal{I}}$ in (2). Note that `ArchAttribute` is independent of input context and thus carries generality over input data instances.

## 3.2  Axioms

We now show how `ArchAttribute` obeys standard feature attribution axioms [47]. Since `ArchAttribute` operates on feature sets, we generalize the notion of standard axioms to feature sets. To this end, we also propose a new axiom, Set Attribution, which allows us to work with feature sets.

Let $\mathcal{S} = \{\mathcal{I}_i\}_{i=1}^s$ be all $s$ feature interactions and main effects of $f$ in the space $\mathcal{X}$ (defined in §2), where we take the union of overlapping sets in $\mathcal{S}$. Later in §4, we explain how to obtain $\mathcal{S}$.

**Completeness:** We consider a generalization of the completeness axiom for which the sum of all attributions equals $f(\mathbf{x}^\star) - f(\mathbf{x}')$. The axiom tells us how much feature(s) impact a prediction.

**Lemma 2** (Completeness on $\mathcal{S}$)**.** *The sum of all attributions by* `ArchAttribute` *for the disjoint sets in $\mathcal{S}$ equals the difference of $f$ between $\mathbf{x}^\star$ and the baseline $\mathbf{x}'$: $f(\mathbf{x}^\star) - f(\mathbf{x}')$.*

The proof is in Appendix C. We can easily see `ArchAttribute` satisfying this axiom in the limiting case where $s = 1$, $\mathcal{I}_1 = \{i\}_{i=1}^p$ because (2) directly becomes $f(\mathbf{x}^\star) - f(\mathbf{x}')$. Existing interaction / group attribution methods: Sampling Contextual Decomposition (SCD) [26], its variant (CD) [36, 43], Sampling Occlusion (SOC) [26], and Shapley Interaction Index (SI) [20] do not satisfy completeness, whereas Integrated Hessians (IH) [25] and Shapley Taylor Interaction Index (STI) [14] do.

**Set Attribution:** We propose an axiom for interaction attribution called **Set Attribution** to work with feature sets as opposed to individual features and follow the additive structure of a function.

**Axiom 3** (Set Attribution)**.** *If $f : \mathbb{R}^p \to \mathbb{R}$ is a function in the form of $f(\mathbf{x}) = \sum_{i=1}^s \varphi_i(\mathbf{x}_{\mathcal{I}_i})$ where $\{\mathcal{I}_i\}_{i=1}^s$ are disjoint and functions $\{\varphi_i(\cdot)\}_{i=1}^s$ have roots, then an interaction attribution method admits an attribution for feature set $\mathcal{I}_i$ as $\varphi_i(\mathbf{x}_{\mathcal{I}_i}) \; \forall i = 1, \ldots, s$.*

For example, if we consider a function $y = x_1 x_2 + x_3$; it makes sense for the attribution of the $x_1 x_2$ interaction to be the value of $x_1 x_2$ and the attribution for the $x_3$ main effect to be the value of $x_3$.

**Lemma 4** (Set Attribution on $\mathcal{S}$)**.** *For $\mathbf{x} = \mathbf{x}^\star$ and a baseline $\mathbf{x}'$ such that $\varphi_i(\mathbf{x}'_{\mathcal{I}_i}) = 0 \; \forall i = 1, \ldots, s$,* `ArchAttribute` *satisfies the Set Attribution axiom and provides attribution $\varphi_i(\mathbf{x}_{\mathcal{I}_i})$ for set $\mathcal{I}_i \; \forall i$.*

The proof is in Appendix E, which follows from Lemma 2. Neither SCD, CD, SOC, SI, IH, nor STI satisfy Set Attribution (shown in Appendix E.1). We can enable Integrated Gradients (IG) [47] to satisfy our axiom by summing its attributions within each feature set of $\mathcal{S}$. `ArchAttribute` differs from IG by its "island effect" (§3.1) and model-agnostic properties.

**Other Axioms:** `ArchAttribute` also satisfies the remaining axioms: Sensitivity, Implementation Invariance, Linearity, and Symmetry-Preserving, which we show via Lemmas 7-11 in Appendix F.

**Discussion:** Several axioms required disjoint interaction and main effect sets in $\mathcal{S}$. Though interactions are not necessarily disjoint by definition (Def. 1), it is reasonable to merge overlapping interactions to obtain compact and simpler visualizations, as shown in Fig. 1 and our experiments later in §5.3. The disjoint sets also allow `ArchAttribute` to yield identifiable non-additive attributions in the sense that it can identify the attribution given a feature set in $\mathcal{S}$. This contrasts with Model-Agnostic Hierarchical Explanations (MAHE) [51], which yields unidentifiable attributions [57].

## 4  `Archipelago` Interaction Detection

Our axiomatic analysis of `ArchAttribute` relied on $\mathcal{S}$, which contains interaction sets of $f$ on the space $\mathcal{X}$ (defined in §2). To develop an interaction detection method that works in tandem with `ArchAttribute`, we draw inspiration from the discrete interpretation of mixed partial derivatives.

### 4.1  Discrete Interpretation of Mixed Partial Derivatives

Consider the plots in Fig. 2, which consist of points $\mathbf{a}$, $\mathbf{b}$, $\mathbf{c}$, and $\mathbf{d}$ that each contain two feature values. From a top-down view of each plot, the points form the corners of a rectangle, whose side lengths are $h_1 = |a_1 - b_1| = |c_1 - d_1|$ and $h_2 = |a_2 - c_2| = |b_2 - d_2|$. When $h_1$ and $h_2$ are small, the mixed partial derivative w.r.t variables $x_1$ and $x_2$ is computed as follows. First, $\dfrac{\partial f(\mathbf{a})}{\partial x_1} \approx \dfrac{1}{h_1} \left( f(\mathbf{a}) - f(\mathbf{b}) \right)$

and $\frac{\partial f(\mathbf{c})}{\partial x_1} \approx \frac{1}{h_1}\left(f(\mathbf{c}) - f(\mathbf{d})\right)$. Similarly, the mixed partial derivative is approximated as:

$$\frac{\partial^2 f}{\partial x_1 x_2} \approx \frac{1}{h_2}\left(\frac{\partial f(\mathbf{a})}{\partial x_1} - \frac{\partial f(\mathbf{c})}{\partial x_1}\right) \approx \frac{1}{h_1 h_2}\left((f(\mathbf{a}) - f(\mathbf{b})) - (f(\mathbf{c}) - f(\mathbf{d}))\right). \tag{3}$$

When $h_1$ and $h_2$ become large, (3) tells us if a plane can fit through all four points $\mathbf{a},\mathbf{b},\mathbf{c},\mathbf{d}$ (Fig. 2a), which occurs when (3) is zero. Here, a plane in the linear form $f(\mathbf{x}) = w_1 x_1 + w_2 x_2 + b$ is functionally equivalent to all functions of the form $f(\mathbf{x}) = f_1(x_1) + f_2(x_2) + b$ since $x_1$ and $x_2$ only take two possible values each, so any deviation from the plane (e.g. Fig. 2b) becomes non-additive. Consequently, a *non-zero* value of (3) identifies a non-additive interaction by the definition of statistical interaction (Def. 1). What's more, the magnitude of (3) tells us the degree of deviation from the plane, or the degree of non-additivity. (Additional details in Appendix G)

## 4.2 ArchDetect

Leveraging these insights about mixed partial derivatives, we now discuss the two components of our proposed interaction detection technique – ArchDetect.

### 4.2.1 Handling Context:
As defined in §3.2 and §4, our problem is how to identify interactions of $p$ features in $\mathcal{X}$ for our input data instance $\mathbf{x}^\star$ and baseline $\mathbf{x}'$. If $p = 2$, then we can almost directly use (3), where $\mathbf{a} = (x_1^\star, x_2^\star)$, $\mathbf{b} = (x_1', x_2^\star)$, $\mathbf{c} = (x_1^\star, x_2')$, and $\mathbf{d} = (x_1', x_2')$. However if $p > 2$, all possible combinations of features in $\mathcal{X}$ would need to be examined to thoroughly identify just one pairwise interaction. To see this, we first rewrite (3) to accommodate $p$ features, and square the result to measure interaction strength and be consistent with previous interaction detectors [18, 19]. The interaction strength between features $i$ and $j$ for a context $\mathbf{x}_{\backslash\{i,j\}}$ is then defined as

$$\omega_{i,j}(\mathbf{x}) = \left(\frac{1}{h_i h_j}\left(f(\mathbf{x}^\star_{\{i,j\}} + \mathbf{x}_{\backslash\{i,j\}}) - f(\mathbf{x}'_{\{i\}} + \mathbf{x}^\star_{\{j\}} + \mathbf{x}_{\backslash\{i,j\}})\right.\right.$$
$$\left.\left. - f(\mathbf{x}^\star_{\{i\}} + \mathbf{x}'_{\{j\}} + \mathbf{x}_{\backslash\{i,j\}}) + f(\mathbf{x}'_{\{i,j\}} + \mathbf{x}_{\backslash\{i,j\}})\right)\right)^2, \tag{4}$$

where $h_i = |x_i^\star - x_i'|$ and $h_j = |x_j^\star - x_j'|$. The thorough way to identify the $\{i,j\}$ feature interaction is given by $\bar{\omega}_{i,j} = \mathbb{E}_{\mathbf{x}\in\mathcal{X}}\left[\omega_{i,j}(\mathbf{x})\right]$, where each element of $\mathbf{x}_{\backslash\{i,j\}}$ is Bernoulli (0.5). This expectation is intractable because $\mathcal{X}$ has an exponential search space, so we propose the first component of ArchDetect for efficient pairwise interaction detection:

$$\bar{\omega}_{i,j} = \frac{1}{2}\left(\omega_{i,j}(\mathbf{x}^\star) + \omega_{i,j}(\mathbf{x}')\right). \tag{5}$$

Here, we estimate the expectation by leveraging the physical meaning of the interactions and ArchAttribute's axioms via the different contexts of $\mathbf{x}$ in (5) as follows:

- **Context of $\mathbf{x}^\star$:** An important interaction is one due to multiple $\mathbf{x}^\star$ features. As a concrete example, consider an image representation of a cat which acts as our input data instance. The following higher-order interaction, *if $x_{ear} = x_{ear}^\star$ and $x_{nose} = x_{nose}^\star$ and $x_{fur} = x_{fur}^\star$ then $f(\mathbf{x}) = high\ cat\ probability$*, is responsible for classifying "cat". We can detect any pairwise subset $\{i,j\}$ of this interaction by setting the context as $\mathbf{x}^\star_{\backslash\{i,j\}}$ using $\omega_{i,j}(\mathbf{x}^\star)$.

- **Context of $\mathbf{x}'$:** Next, we consider $\mathbf{x}'_{\backslash\{i,j\}}$ to detect interactions via $\omega_{i,j}(\mathbf{x}')$, which helps us establish ArchAttribute's completeness (Lemma 2). This also separates out effects of any higher-order baseline interactions from $f(\mathbf{x}')$ in (8) (Appendix C) and recombine their effects in (11). From an interpretability standpoint, the $\mathbf{x}'_{\backslash\{i,j\}}$ context ranks pairwise interactions w.r.t. a standard baseline. This context is also used by ArchAttribute (2).

- **Other Contexts:** The first two contexts accounted for any-order interactions created by either input or baseline features and a few interactions created by a mix of input and baseline features. The remaining interactions specifically require a mix of $> 3$ input and baseline features. This case is unlikely and is excluded, as we discuss next.

The following assumption formalizes our intuition for the *Other Contexts* setting where there is a mix of higher-order ($> 3$) input and baseline feature interactions.

Table 1: Comparison of interaction detectors (b) on synthetic ground truth in (a).

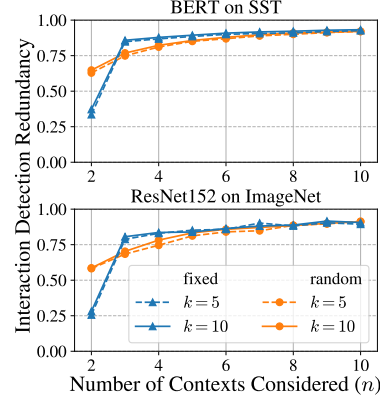

(a) Functions with Ground Truth Interactions

$$F_1(\mathbf{x}) = \sum_{i=1}^{10}\sum_{j=1}^{10} x_i x_j + \sum_{i=11}^{20}\sum_{j=21}^{30} x_i x_j + \sum_{k=1}^{40} x_k$$
$$F_2(\mathbf{x}) = \bigwedge(\mathbf{x}; \{x_i^\star\}_{i=1}^{20}) + \bigwedge(\mathbf{x}; \{x_i^\star\}_{i=11}^{30}) + \sum_{j=1}^{40} x_j$$
$$F_3(\mathbf{x}) = \bigwedge(\mathbf{x}; \{x_i'\}_{i=1}^{20}) + \bigwedge(\mathbf{x}; \{x_i^\star\}_{i=11}^{30}) + \sum_{j=1}^{40} x_j$$
$$F_4(\mathbf{x}) = \bigwedge(\mathbf{x}; \{x_1^\star, x_2^\star\} \cup \{x_3'\}) + \bigwedge(\mathbf{x}; \{x_i^\star\}_{i=11}^{30}) + \sum_{j=1}^{40} x_j$$

(b) Pairwise Interaction Ranking AUC. The baseline methods often fail to detect interactions suited for the desired contexts in §4.2.1.

| Method | $F_1$ | $F_2$ | $F_3$ | $F_4$ |
|---|---|---|---|---|
| Two-way ANOVA | 1.0 | 0.51 | 0.51 | 0.55 |
| Integrated Hessians | 1.0 | N/A | N/A | N/A |
| Neural Interaction Detection | 0.94 | 0.54 | 0.54 | 0.56 |
| Shapley Interaction Index | 1.0 | 0.50 | 0.50 | 0.51 |
| Shapley Taylor Interaction Index | 1.0 | 0.55 | 0.78 | 0.55 |
| ArchDetect (this work) | 1.0 | 1.0 | 1.0 | 1.0 |

Figure 3: Interaction detection overlap (redundancy) with added contexts to (5). "fixed" at $n = 2$ (ArchDetect) already shows good stability.

**Assumption 5** (Higher-Order Mixed-Interaction). *For any feature set $\mathcal{I}$ where $|\mathcal{I}| > 3$ and any pair of non-empty disjoint sets $\mathcal{A}$ and $\mathcal{B}$ where $\mathcal{A} \cup \mathcal{B} = \mathcal{I}$, the instances $\mathbf{x} \in \mathcal{X}$ such that $x_i = x_i^\star \; \forall i \in \mathcal{A}$ and $x_j = x_j' \; \forall j \in \mathcal{B}$ do not cause a higher-order interaction of all features $\{x_k\}_{k \in \mathcal{I}}$ via $f$.*

Assumption 5 has a similar intuition as ArchAttribute in §3.1 that input feature values do not specifically interact with baseline feature values. To understand this assumption, consider the original sentiment analysis example in Fig. 1 simplified as $\mathbf{x}^\star$ = "bad terrible awful horrible movie" where $\mathbf{x}'$ = "_ _ _ _ _". It is reasonable to assume that there is no special interaction created by token sets such as {bad, terrible, _ , horrible} or {_ , _ , _ , horrible} due to the meaningless nature of the "_" token. In practice, it makes more sense to detect interactions out of sets like {bad, terrible, horrible} so we can avoid spurious interactions with the "_" token.

**Efficiency:** In (5), ArchDetect attains interaction detection over all pairs $\{i, j\}$ in $\mathcal{O}(p^2)$ calls of $f$. Note that in (4), most function calls are reusable during pairwise interaction detection.

**4.2.2 Detecting Disjoint Interaction Sets:** In this section, the aim here is to recover arbitrary size and disjoint non-additive feature sets $\mathcal{S} = \{\mathcal{I}_i\}$ (not just pairs). ArchDetect looks at the union of overlapping pairwise interactions to obtain disjoint feature sets. Merging these pairwise interactions captures any existing higher-order interactions automatically since the existence of a higher-order interaction automatically means all its subset interactions exist (§2). In addition, ArchDetect merges these overlapped pairwise interactions with all individual feature effects to account for all features. Note that to visualize ArchDetect, an initial threshold on the pairwise interaction ranking of (5) may be needed prior to merging, to show smaller size interactions for explanation simplicity.

## 5 Experiments

### 5.1 Setup

We conduct experiments first on ArchDetect in §5.2 then on ArchAttribute in §5.3. We then visualize their combined form as Archipelago in §5.3. Throughout our experiments, we commonly study BERT [13, 56] on text-based sentiment analysis and ResNet152 [24] on image classification. BERT was fine-tuned on the SST dataset [44], and ResNet152 was pretrained on ImageNet [12].

For sentiment analysis, we set the baseline vector $\mathbf{x}'$ to be the tokens "_", in place of each word-token from $\mathbf{x}^\star$. For image classification, we set $\mathbf{x}'$ to be an all-zero image, and use the Quickshift superpixel segmenter [53] as per the need for input dimensionality reduction [48] (details in Appendix B). We set $h_i = h_j = 1$ for both domains. Several methods we compare to are common across experiments, in particular IG, IH, (disjoint) MAHE, SI, STI, and Difference, defined as $\phi_d(\mathcal{I}) = f(\mathbf{x}^\star) - f(\mathbf{x}'_\mathcal{I} + \mathbf{x}^\star_{\backslash \mathcal{I}})$.

### 5.2 ArchDetect

We validate ArchDetect's performance via synthetic ground truth and redundancy experiments.

Table 2: Comparison of attribution methods on BERT for sentiment analysis and ResNet152 for image classification. Performance is measured by the correlation ($\rho$) or AUC of the top and bottom 10% of attributions for each method with respect to reference scores defined in §5.3.

| Method | BERT Sentiment Analysis | | ResNet152 Image Classification |
| --- | --- | --- | --- |
| | Word $\rho$ | Phrase $\rho$ † | Segment AUC † |
| Difference | 0.333 | 0.639 | 0.705 |
| Integrated Gradients (IG) | 0.473 | 0.737 | 0.786 |
| Integrated Hessians (IH) | N/A | 0.128 | N/A |
| Model-Agnostic Hierarchical Explanations (MAHE) | 0.570 | 0.702 | 0.712 |
| Shapley Interaction Index (SI) | 0.160 | −0.018 | 0.530 |
| Shapley Taylor Interaction Index (STI) | 0.657 | 0.286 | 0.626 |
| *Sampling Contextual Decomposition (SCD) | 0.622 | 0.742 | N/A |
| *Sampling Occlusion (SOC) | 0.670 | 0.794 | N/A |
| `ArchAttribute` (this work) | **0.745** | **0.836** | **0.919** |

† Methods that cannot tractably run for arbitrary feature set sizes are only run for pairwise feature sets.
* SCD and SOC are specifically for sequence models and contiguous words.

**Synthetic Validation:** We set $\mathbf{x}^\star = [1, 1, \ldots, 1] \in \mathbb{R}^{40}$ and $\mathbf{x}' = [-1, -1, \ldots, -1] \in \mathbb{R}^{40}$. Let $z[\cdot]$ be a key-value pair function such that $z[i] = x_i$ for key $i \in z.keys$ and value $x_i$, so we can define

$$\bigwedge(\mathbf{x}; z) := \begin{cases} 1, & \text{if } x_i = z[i] \ \forall i \in z.keys \\ -1 & \text{for all other cases.} \end{cases}$$

Table 1a shows functions with ground truth interactions suited for the desired contexts in §4.2.1. Table 1b shows interaction detection AUC on these functions by `ArchDetect`, IH, SI, STI, Two-way ANOVA [16] and the state-of-the-art Neural Interaction Detection [49]. On $F_2$, $F_3$, & $F_4$, the baseline methods often fail because they are not designed to detect the desired interactions from §4.2.1.

**Interaction Redundancy:** The purpose of the next experiment is to see if `ArchDetect` can omit certain higher-order interactions. We study the form of (5) by examining the redundancy of interactions as new contexts are added to (5), which we now write as $\bar{\omega}_{i,j}(C) = \frac{1}{C} \sum_{c=1}^{C} \omega_{i,j}(\mathbf{x}_c)$. Let $n$ be the number of contexts considered, and $k$ be the number of top pairwise interactions selected after running pairwise interaction detection via $\bar{\omega}_{i,j}$ for all $\{i, j\}$ pairs. Interaction redundancy is the overlap ratio of two sets of top-$k$ pairwise interactions, one generated via $\bar{\omega}_{i,j}(n)$ and the other one via $\bar{\omega}_{i,j}(n-1)$ for some integer $n \geq 2$. We generally expect the redundancy to increase as $n$ increases, which we initially observe in Fig. 3. Here, "fixed" and "random" correspond to different context sequences $\mathbf{x}_1, \mathbf{x}_2, \ldots, \mathbf{x}_N$. The "random" sequence uses random samples from $\mathcal{X}$ for all $\{\mathbf{x}_i\}_{i=1}^N$, whereas the "fixed" sequence is fixed in the sense that $\mathbf{x}_1 = \mathbf{x}^\star$, $\mathbf{x}_2 = \mathbf{x}'$, and the remaining $\{\mathbf{x}_i\}_{i=3}^N$ are random samples. Experiments are done on the SST test set for BERT and 100 random test images in ImageNet for ResNet152. Notably, the "fixed" setting has very low redundancy at $n = 2$ (`ArchDetect`) versus "random". As soon as $n = 3$, the redundancy jumps and stabilizes quickly. These experiments support Assumption 5 and (5) to omit specified higher-order interactions.

### 5.3 `ArchAttribute` & `Archipelago`

We study the coherent interpretability of `ArchAttribute` by comparing its attribution scores to ground truth annotations on subsets of features. For fair comparison, we look at extreme attributions (top and bottom 10%) for each baseline method. We then visualize the combined `Archipelago` framework. Additional comparisons on attributions and visualizations are shown in Appendices I, J.

**Sentiment Analysis:** For this task, we compare `ArchAttribute` to other explanation methods on two metrics: phrase correlation (Phrase $\rho$) and word correlation (Word $\rho$) on the SST test set. Phrase $\rho$ is the Pearson correlation between estimated phrase attributions and SST phrase labels (excluding prediction labels) on a 5-point sentiment scale [26]. Word $\rho$ is the same as Phrase $\rho$ but only for single words that consist of a single token. In addition to the aforementioned baseline methods in §5.1, we include the state-of-the-art SCD and SOC methods for sequence models [26] in our evaluation. In Table 2, `ArchAttribute` compares favorably to all methods where we consider the top and bottom 10% of the attribution scores for each method. We obtain similar performance across all other percentiles in Appendix I.

We visualize `Archipelago` explanations on $\mathcal{S}$ generated by top-3 pairwise interactions (§4.2.2) in Fig. 4. The sentence examples are randomly selected from the SST test set. The visualizations

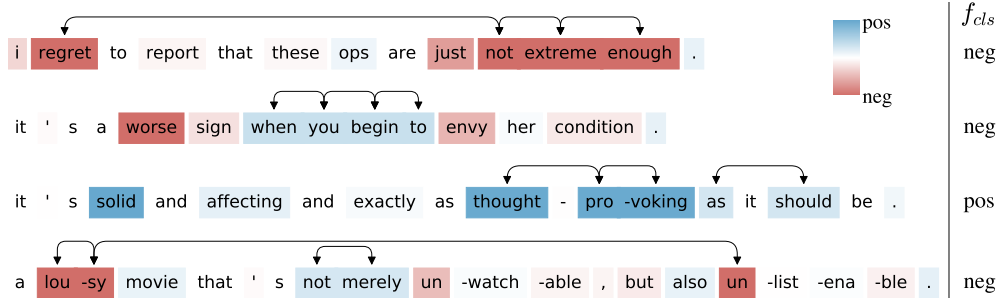

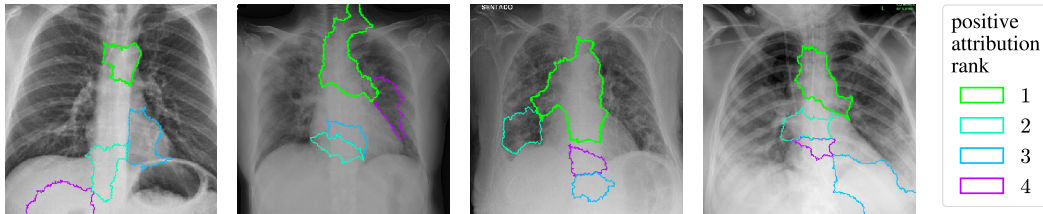

Figure 4: Our BERT visualizations on random test sentences from SST under BERT tokenization. Arrows indicate interactions, and colors indicate attribution scores. $f_{cls}$ is the sentiment classification. The interactions point to salient and sometimes long-range sets of words, and the colors are sensible.

Figure 5: Our explanations of a COVID-19 classifier (COVID-Net) [54] on randomly selected test X-rays [9, 10] classified as COVID positive. COVID-Net accurately distinguishes COVID from pneumonia and normal X-rays. Colored outlines indicate detected feature sets with positive attribution. The interactions consistently focus on the "great vessel" region outlined in green.

show interactions and individual feature effects which all have reasonable polarity and intensity. Interestingly, some of the interactions, e.g. between "lou-sy" and "un", are long range.

**Image Classification:** On image classification, we compare `ArchAttribute` to relevant baseline methods on a "Segment AUC" metric, which computes the agreement between the estimated attribution of an image segment and that segment's label. We obtain segment labels from the MS COCO dataset [29] and match them to the label space of ImageNet. All explanation attributions are computed relative to ResNet152's top-classification in the joint label space. The segment label thus becomes whether or not the segment belongs to the same class as the top-classification. Evaluation is conducted on all segments with valid labels in the MS COCO dev set. `ArchAttribute` performs especially well on extreme attributions in Table 2, as well as all attributions (in Appendix I).

Fig. 5 visualizes `Archipelago` on an accurate coronavirus (COVID-19) classifier for chest X-rays [54], where $\mathcal{S}$ is generated by top-5 pairwise interactions (§4.2.2). Shown is a random selection of test X-rays [9,10] that are classified COVID-positive. The explanations tend to detect the "great vessels" near the heart.

**Recommendation Task:** Fig. 6 shows `Archipelago`'s result for this task using a state-of-the-art AutoInt model [45] for ad-recommendation. Here, our approach finds a positive interaction between "device_id" and "banner_pos" in the Avazu dataset [1], meaning that the online advertisement model decides the banner position based on user device_id. Note that for this task, there are no ground truth annotations.

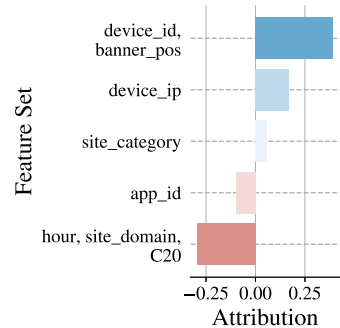

Figure 6: Online ad-targeting: "banner_pos" is used to target ads to a user per their "device_id".

### 5.4 Interactive Visualization

While our visualizations have used a fixed threshold on pairwise interaction strength, we can interactively visualize `Archipelago` explanations by varying the threshold. For example, a slider user interface offers this interactivity and allows users to perform in-depth analysis. Fig. 7 illustrates the interface, where moving the slider tells us when interactions appear and allows us to better judge

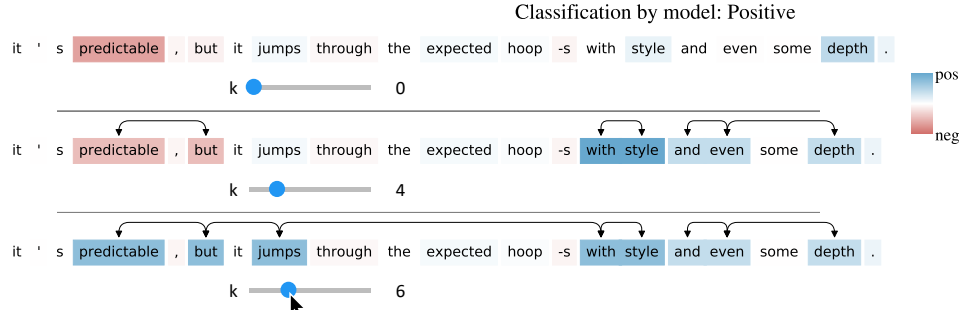

Figure 7: Interactive visualization of `Archipelago`. When moving the slider, the initial negativity of "predictable" and "but" turns positive after interacting with the positive phrase "jumps with style".

model quality. Note that our interactive visualization is fast since interaction detection only runs once, and the additional `ArchAttribute` and interaction-merge steps are fast.

## 5.5 Runtime

Fig. 8 shows a serial runtime comparison of explainer methods for (a) BERT sentiment analysis on SST and (b) ResNet152 image classification on ImageNet. Runtimes correspond to static explanations and are averaged across 100 random data samples from respective test sets. `Archipelago` outperforms the state-of-the-art. These experiments were done on a server with 32 Intel Xeon E5-2640 v2 CPUs @ 2.00GHz and 2 Nvidia 1080 Ti GPUs.

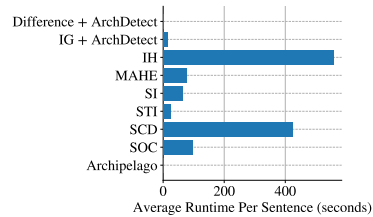

(a) Explaining Sentiment Analysis

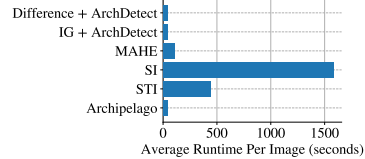

(b) Explaining Image Classification

Figure 8: Runtime Comparison

## 6 Related Works

**Attribution:** Individual feature attribution methods distill any interactions of a data instance as attribution scores for each feature. Many methods require the scores to sum to equal the output [7, 32, 39, 41, 47], such as LIME and SHAP, which train surrogate linear explainer models on feature perturbations, and IG which invokes the fundamental theorem of calculus. Other methods compute attributions from an information theoretic perspective [8] or strictly from model gradients [4, 40, 42]. These methods interpret feature importance but not feature interactions.

**Feature Interaction:** Feature interaction explanation methods tend to either perform interaction detection [2, 6, 16, 18, 19, 46, 49] or combined interaction detection and attribution [14, 25, 30, 31, 38, 51]. Relevant black-box interaction explainers are STI [14] which uses random feature orderings to identify contexts for a variant of (4) so that interaction scores satisfy completeness, IH [25] which extends IG with path integration for hessian computations, and MAHE [51] which trains surrogate explainer models for interaction detection and attribution. STI and IH are axiomatic and satisfy completeness but their attributions are uninterpretable (Table 2) and inefficient. MAHE's attributions are unidentifiable by training additive attribution models on overlapping feature sets. Several methods compute attributions on feature sequences or sets, such as SOC [26], SCD [26], and CD [36, 43], but they do not obey basic axioms. Finally, many methods are not model-agnostic, such as SCD, CD, IG, IH, GA2M [30], and Tree-SHAP [31]. Additional earlier works are discussed in Appendix H.

## 7 Discussion

Understandable and accessible explanations are cornerstones of interpretability which informed our design of `Archipelago`. Here, we develop an interpretable, model-agnostic, axiomatic, and efficient interaction explainer which achieves state-of-the-art results on multiple attribution tasks and offers compelling visualizations of model predictions. In addition, we introduce a new axiom and generalize existing axioms to higher-order interaction settings. This provides guidance on how to design interaction attribution methods. To be able to solve the transparency issue, we need to understand feature attribution better. This work proposes interpretable and axiomatic feature interaction explanations to motivate future explorations in this area.

## Broader Impact

The purpose of this work is to provide new insights into existing and future prediction models. The explanations from `Archipelago` can be used by both machine learning practitioners and audiences without background expertise. The societal risk of this work is any overdependence on `Archipelago`. Users of this explanation method should consider the merits of not only this method but also other explanation methods for their use cases. For example, users may want fine-grained pixel-level explanations of image classifications whereas our explanations may require superpixel segmentation. Nevertheless, we believe this work can help reveal biases in prediction models, assist in scientific discovery, and stimulate discussions on how to debug models based on feature interactions.

## Acknowledgments and Disclosure of Funding

We sincerely thank our anonymous reviewers for their invaluable feedback.
Funding in direct support of this work: National Science Foundation Awards IIS-1254206 and IIS-1539608.

## Footnotes

[1]Code is available at: https://github.com/mtsang/archipelago

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
