[Supplementary Material]

# Appendix

## A Acronyms

Table 3: Acronym Definitions

| Acronym | Meaning |
|---------|---------|
| pos | positive |
| neg | negative |
| IG | Integrated Gradients [47] |
| IH | Integrated Hessians [25] |
| MAHE | Model-Agnostic Hierarchical Explanations [51] |
| SI | Shapley Interaction Index [20] |
| STI | Shapley Taylor Interaction Index [14] |
| SCD | Sampling Contextual Decomposition [26] |
| SOC | Sampling Occlusion [26] |
| ANOVA | Analysis of Variance [16] |
| LIME | Locally Interpretable Model-Agnostic Explanations [39] |
| SHAP | Shapley Additive Explanations [32] |
| GA2M | Generalized Additive Model with Pairwise Interactions [30] |
| MS COCO | Microsoft Common Objects in Context [29] |
| SST | Stanford Sentiment Treebank [44] |
| BERT | Bidirectional Encoder Representations from Transformers [13] |
| AUC | Area Under the Receiver Operating Characteristic Curve |
| COVID | Coronavirus Disease |

## B Input Dimensionality Reduction

For a black-box model $f : \mathbb{R}^{p'} \to \mathbb{R}$ which takes as input a vector with $p'$ dimensions (e.g. an image, input embedding, etc.) and maps it to a scalar output (e.g. a class logit), we can make `ArchDetect` more efficient by operating on a lower dimensional input encoding $\mathbf{x} \in \mathbb{R}^p$ with $p$ dimensions. To match the dimensionality $p'$ of the input argument of $f$, we define a transformation function $\xi : \mathbb{R}^p \to \mathbb{R}^{p'}$ which takes the input encoding $\mathbf{x}$ in the lower dimensional space $p$ and brings it back to the input space of $f$ with dimensionality $p'$. In other words, (4) becomes

$$\omega_{i,j}(\mathbf{x}) = \left( \frac{1}{h_i h_j} \left( f'(\mathbf{x}^\star_{\{i,j\}} + \mathbf{x}_{\backslash\{i,j\}}) - f'(\mathbf{x}'_{\{i\}} + \mathbf{x}^\star_{\{j\}} + \mathbf{x}_{\backslash\{i,j\}}) - f'(\mathbf{x}^\star_{\{i\}} + \mathbf{x}'_{\{j\}} + \mathbf{x}_{\backslash\{i,j\}}) \right. \right.$$
$$\left. \left. + f'(\mathbf{x}'_{\{i,j\}} + \mathbf{x}_{\backslash\{i,j\}}) \right) \right)^2,$$

where $f' = f \circ \xi$. Correspondingly, `ArchAttribute` (2) becomes

$$\phi(\mathcal{I}) = f'(\mathbf{x}^\star_\mathcal{I} + \mathbf{x}'_{\backslash\mathcal{I}}) - f'(\mathbf{x}').$$

Examples of input encodings are discussed for the following data types:

- For an image, we use a superpixel segmenter, which selects regions on the image. The selection is covered by the vector $\mathbf{x} \in \{0,1\}^p$, which encodes which image segments have been selected. Note that wherever $\mathbf{x}$ is 0 corresponds to a baseline feature value (e.g. zeroed image pixels).

- For text, we use the natural correspondence between an input embedding and a word token. The selection of input embedding vectors is also covered by the vector $\mathbf{x} \in \{0,1\}^p$.

- For recommendation data, we use the same type of correspondence between an input embedding and a feature field.

Similar notions of input encodings have also been used in [39, 48].

## C Completeness Axiom

**Lemma 2** (Completeness on $\mathcal{S}$). *The sum of all attributions by* `ArchAttribute` *for the disjoint sets in $\mathcal{S}$ equals the difference of $f$ between $\mathbf{x}^\star$ and the baseline $\mathbf{x}'$: $f(\mathbf{x}^\star) - f(\mathbf{x}')$.*

*Proof.* Based on the definition of non-additive statistical interaction (Def. 1), a function $f$ can be represented as a generalized additive function [49–51], here on the domain of $\mathcal{X}$:

$$f(\mathbf{x}) = \sum_{i=1}^{\eta} q_i(\mathbf{x}_{\mathcal{I}_i^u}) + \sum_{j=1}^{p} q_j'(x_j) + b, \tag{6}$$

where $q_i(\mathbf{x}_{\mathcal{I}_i^u})$ is a function of each interaction $\mathcal{I}_i^u$ on $\mathcal{X}$ $\forall i = 1, \ldots, \eta$ interactions, $q_j'(x_j)$ is a function for each feature $\forall j = 1, \ldots, p$, and $b$ is a bias. The $u$ in $\mathcal{I}^u$ stands for "unmerged".

The disjoint sets of $\mathcal{S} = \{\mathcal{I}_i\}_{i=1}^{s}$ are the result of merging overlapping interaction sets and main effect sets, so we can merge the subfunctions $q(\cdot)$ and $q'(\cdot)$ of (6) whose input sets overlap to write $f(\mathbf{x})$ as a sum of new functions $g_i(\mathbf{x}_{\mathcal{I}_i})$ $\forall i = 1, \ldots, s$:

$$f(\mathbf{x}) = \sum_{i=1}^{s} g_i(\mathbf{x}_{\mathcal{I}_i}) + b. \tag{7}$$

For some $\{g_i\}_{i=1}^{s}$ of the form of (7), we rewrite (2) by separating out the effect of index $i$:

$$\phi(\mathcal{I}_i) = f(\mathbf{x}_{\mathcal{I}_i}^\star + \mathbf{x}_{\backslash\mathcal{I}_i}') - f(\mathbf{x}') \quad \forall i = 1, \ldots, s$$

$$= \left( g_i(\mathbf{x}_{\mathcal{I}_i}^\star) + \sum_{\substack{j=1 \\ j \neq i}}^{s} g_j(\mathbf{x}_{\mathcal{I}_j}') + b \right) - \left( g_i(\mathbf{x}_{\mathcal{I}_i}') + \sum_{\substack{j=1 \\ j \neq i}}^{s} g_j(\mathbf{x}_{\mathcal{I}_j}') + b \right) \tag{8}$$

$$= g_i(\mathbf{x}_{\mathcal{I}_i}^\star) - g_i(\mathbf{x}_{\mathcal{I}_i}'). \tag{9}$$

Since all $\mathcal{I} \in \mathcal{S}$ are disjoint, $g_j(\mathbf{x}_{\mathcal{I}_j}')$ can be canceled in (8) $\forall j$, leading to (9). The result at (9) can also be obtained with an alternative attribution approach, as shown in Corollary 6.

Next, we compute the sum of attributions:

$$\sum_{i=1}^{s} \phi(\mathcal{I}_i) = \sum_{i=1}^{s} \left( g_i(\mathbf{x}_{\mathcal{I}_i}^\star) - g_i(\mathbf{x}_{\mathcal{I}_i}') \right) \tag{10}$$

$$= \sum_{i=1}^{s} g_i(\mathbf{x}_{\mathcal{I}_i}^\star) - \sum_{i=1}^{s} g_i(\mathbf{x}_{\mathcal{I}_i}') \tag{11}$$

$$= f(\mathbf{x}^\star) - f(\mathbf{x}')$$

$\square$

## D Completeness of a Complementary Attribution Method

**Corollary 6** (Completeness of a Complement). *An attribution approach: $\phi(\mathcal{I}) = f(\mathbf{x}^\star) - f(\mathbf{x}_{\mathcal{I}}' + \mathbf{x}_{\backslash\mathcal{I}}^\star)$, similar to what is mentioned in [26, 28], also satisfies the completeness axiom.*

*Proof.* Based on Eqs. 7 - 9 of Lemma 2:

$$\phi(\mathcal{I}_i) = f(\mathbf{x}^\star) - f(\mathbf{x}_{\mathcal{I}_i}' + \mathbf{x}_{\backslash\mathcal{I}_i}^\star)$$

$$= \left( g_i(\mathbf{x}_{\mathcal{I}_i}^\star) + \sum_{\substack{j=1 \\ j \neq i}}^{s} g_j(\mathbf{x}_{\mathcal{I}_j}^\star) + b \right) - \left( g_i(\mathbf{x}_{\mathcal{I}_i}') + \sum_{\substack{j=1 \\ j \neq i}}^{s} g_j(\mathbf{x}_{\mathcal{I}_j}^\star) + b \right)$$

$$= g_i(\mathbf{x}_{\mathcal{I}_i}^\star) - g_i(\mathbf{x}_{\mathcal{I}_i}')$$

We can then resume with (10) of Lemma 2. $\square$

# E    Set Attribution Axiom

**Axiom 3** (Set Attribution). *If $f : \mathbb{R}^p \to \mathbb{R}$ is a function in the form of $f(\mathbf{x}) = \sum_{i=1}^s \varphi_i(\mathbf{x}_{\mathcal{I}_i})$ where $\{\mathcal{I}_i\}_{i=1}^s$ are disjoint and functions $\{\varphi_i(\cdot)\}_{i=1}^s$ have roots, then an interaction attribution method admits an attribution for feature set $\mathcal{I}_i$ as $\varphi_i(\mathbf{x}_{\mathcal{I}_i}) \ \forall i = 1, \ldots, s.$*

**Lemma 4** (Set Attribution on $\mathcal{S}$). *For $\mathbf{x} = \mathbf{x}^\star$ and a baseline $\mathbf{x}'$ such that $\varphi_i(\mathbf{x}'_{\mathcal{I}_i}) = 0 \ \forall i = 1, \ldots, s,$* `ArchAttribute` *satisfies the Set Attribution axiom and provides attribution $\varphi_i(\mathbf{x}_{\mathcal{I}_i})$ for set $\mathcal{I}_i \ \forall i.$*

*Proof.* From (9) in Lemma 2, `ArchAttribute` can be written as

$$\phi(\mathcal{I}_i) = g_i(\mathbf{x}^\star_{\mathcal{I}_i}) - g_i(\mathbf{x}'_{\mathcal{I}_i}) \quad \forall i = 1, \ldots, s,$$

where $f(\mathbf{x}) = \sum_{i=1}^s g_i(\mathbf{x}_{\mathcal{I}_i}) + b$. Since $\mathcal{S} = \{\mathcal{I}_i\}_{i=1}^s$ are disjoint feature sets for the same function $f$ in Axiom 3, $g_i(\cdot)$ and $\varphi_i(\cdot)$ are related by a constant bias $b_i$:

$$\varphi_i(\mathbf{x}) = g_i(\mathbf{x}) + b_i$$

Each $\varphi_i(\cdot)$ has roots, so $g_i(\mathbf{x}) + b_i$ has roots. $\mathbf{x}'$ is set such that $\varphi_i(\mathbf{x}'_{\mathcal{I}_i}) = g_i(\mathbf{x}'_{\mathcal{I}_i}) + b_i = 0$. Rearranging,

$$-g_i(\mathbf{x}'_{\mathcal{I}_i}) = b_i.$$

Adding $g_i(\mathbf{x}^\star_{\mathcal{I}_i})$ to both sides,

$$g_i(\mathbf{x}^\star_{\mathcal{I}_i}) - g_i(\mathbf{x}'_{\mathcal{I}_i}) = g_i(\mathbf{x}^\star_{\mathcal{I}_i}) + b_i,$$

which becomes

$$\phi(\mathcal{I}_i) = \varphi_i(\mathbf{x}^\star_{\mathcal{I}_i}) \quad \forall i = 1, \ldots, s.$$

$\square$

## E.1    Set Attribution Counterexamples

We now provide counterexamples to identify situations in which the related methods do not satisfy the Set Attribution axiom.

Let

$$f(\mathbf{x}) = \mathrm{ReLU}(x_1 + x_3 + 1) + \mathrm{ReLU}(x_2) + 1.$$

$f(\mathbf{x})$ can be written as $f(\mathbf{x}) = \varphi_1(\mathbf{x}_{\{1,3\}}) + \varphi_2(\mathbf{x}_{\{2\}})$ where $\varphi_1(\mathbf{x}) = \mathrm{ReLU}(x_1 + x_3 + 1)$, and $\varphi_2(\mathbf{x}) = \mathrm{ReLU}(x_2) + 1$. According to the Set Attribution axiom, an interaction attribution method admits attributions as

- $\mathrm{ReLU}(x_1 + x_3 + 1)$ for features $\mathcal{I}_1 = \{1, 3\}$
- $\mathrm{ReLU}(x_2) + 1$ for feature $\mathcal{I}_2 = \{2\}$.

The above setting serves as counterexamples to the related methods as follows:

- CD always assigns $\alpha + \frac{\alpha}{\alpha + \beta}$ to $\mathcal{I}_1$ and $\beta + \frac{\beta}{\alpha + \beta}$ to $\mathcal{I}_2$, where $\alpha = \mathrm{ReLU}(x_1 + x_3 + 1)$ and $\beta = \mathrm{ReLU}(x_2)$.

- SCD uses an expectation over an activation decomposition, which does not guarantee admission of $\mathrm{ReLU}(x_1 + x_3 + 1)$ for $\mathcal{I}_1$ and $\mathrm{ReLU}(x_2)$ for $\mathcal{I}_2$ through their respective decompositions. In the ideal case SCD becomes CD, which still does not satisfy Set Attribution from above.

- IH always assigns a zero attribution to $\mathcal{I}_2$ from hessian computations. IH also does not assign attributions to general sets of features.

- SOC does not assign attributions to general feature sets, only contiguous feature sequences.

- Both SI and STI assign the following attribution score to $\mathcal{I}_1$:

$$\mathrm{ReLU}(x_1 + x_3 + 1) - \mathrm{ReLU}(x_1 + x'_3 + 1) - \mathrm{ReLU}(x'_1 + x_3 + 1) + \mathrm{ReLU}(x'_1 + x'_3 + 1). \tag{12}$$

  There do not exist a selection of $x'_1$ and $x'_3$ such that this attribution becomes $\mathrm{ReLU}(x_1 + x_3 + 1)$ for all values of $x_1$ and $x_3$.

*Proof.* We prove via case-by-case contradiction. Only the $\mathrm{ReLU}(x_1 + x_3 + 1)$ term can create an interaction between $x_1$ and $x_3$, and this term is also the target result, so any nonzero deviation from this term via independent $x_1$ or $x_3$ effects in (12) must be countered. These independent effects manifest as the $\mathrm{ReLU}(x_1 + x_3' + 1)$ or $\mathrm{ReLU}(x_1' + x_3 + 1)$ terms respectively. Since ReLU is always non-negative, the only way either of these terms is nonzero is if it is positive, which implies that $\mathrm{ReLU}(x_1 + x_3' + 1) = x_1 + x_3' + 1$ or $\mathrm{ReLU}(x_1' + x_3 + 1) = x_1' + x_3 + 1$. If both terms are positive, their substitution into (12) yields $\mathrm{ReLU}(x_1 + x_3 + 1) - x_1 - x_3' - 1 - x_1' - x_3 - 1 + \mathrm{ReLU}(x_1' + x_3' + 1)$. Even if $\mathrm{ReLU}(x_1' + x_3' + 1)$ is positive, we obtain $\mathrm{ReLU}(x_1 + x_3 + 1) - x_1 - x_3' - 1 - x_1' - x_3 - 1 + x_1' + x_3' + 1 = \mathrm{ReLU}(x_1 + x_3 + 1) - x_1 - x_3 - 1$. Asserting $-x_1 - x_3 - 1 = 0$ is a contradiction. If only one of the independent effects was positive, we also cannot assert $0$ through similar simplifications.

Now consider the remaining case where $\mathrm{ReLU}(x_1 + x_3' + 1) = \mathrm{ReLU}(x_1' + x_3 + 1) = \mathrm{ReLU}(x_1' + x_3' + 1) = 0$. For any real-valued $x_1'$ or $x_3'$, there can also be a negative real-valued $x_3$ or $x_1$ respectively. From either terms $\mathrm{ReLU}(x_1 + x_3' + 1)$ or $\mathrm{ReLU}(x_1' + x_3 + 1)$, we obtain $\mathrm{ReLU}(1) = 0$, which is a contradiction. □

# F  Other Axioms

## F.1  Sensitivity

**Lemma 7** (Sensitivity (a)). *If $\mathbf{x}^\star$ and $\mathbf{x}'$ only differ at features indexed in $\mathcal{I}$ and $f(\mathbf{x}^\star) \neq f(\mathbf{x}')$, then $\phi(\mathcal{I})$ (2) yields a nonzero attribution.*

*Proof.* Since $\mathbf{x}^\star$ and $\mathbf{x}'$ only differ at $\mathcal{I}$, the following is true: $\mathbf{x}^\star_{\backslash \mathcal{I}} = \mathbf{x}'_{\backslash \mathcal{I}}$. We can therefore write $\mathbf{x}^\star$ as

$$\mathbf{x}^\star = \mathbf{x}^\star_\mathcal{I} + \mathbf{x}^\star_{\backslash \mathcal{I}}$$
$$= \mathbf{x}^\star_\mathcal{I} + \mathbf{x}'_{\backslash \mathcal{I}}$$

Substituting this equivalence in (2), we have

$$\phi(\mathcal{I}) = f(\mathbf{x}^\star_\mathcal{I} + \mathbf{x}'_{\backslash \mathcal{I}}) - f(\mathbf{x}')$$
$$= f(\mathbf{x}^\star) - f(\mathbf{x}').$$

Since $f(\mathbf{x}^\star) - f(\mathbf{x}') \neq 0$, we directly obtain $\phi(\mathcal{I}) \neq 0$.

□

**Lemma 8** (Sensitivity (b)). *If $f$ does not functionally depend on $\mathcal{I}$, then $\phi(\mathcal{I})$ is always zero.*

*Proof.* Since $f$ does not functionally depend on $\mathcal{I}$,

$$f(\mathbf{x}^\star_\mathcal{I} + \mathbf{x}'_{\backslash \mathcal{I}}) = f(\mathbf{x}'_\mathcal{I} + \mathbf{x}'_{\backslash \mathcal{I}})$$
$$= f(\mathbf{x}')$$

Therefore,

$$\phi(\mathcal{I}) = f(\mathbf{x}^\star_\mathcal{I} + \mathbf{x}'_{\backslash \mathcal{I}}) - f(\mathbf{x}') = 0.$$

□

## F.2  Implementation Invariance

**Lemma 9** (Implementation Invariance). *For functionally equivalent models (with the same input-output mapping), $\phi(\cdot)$ are the same.*

The definition of (2) only relies on function calls to $f$, which implies Implementation Invariance.

## F.3 Linearity

**Lemma 10** (Linearity on $\mathcal{S}$). *If two models $f_1$, $f_2$ have the same disjoint feature sets $\mathcal{S}$ and $f = c_1 f_1 + c_2 f_2$ where $c_1, c_2$ are constants, then $\phi(\mathcal{I}) = c_1 \phi_1(\mathcal{I}) + c_2 \phi_2(\mathcal{I}) \ \forall \mathcal{I} \in \mathcal{S}$.*

*Proof.* Since $f_1$ and $f_2$ have the same $\mathcal{S} = \{\mathcal{I}_i\}_{i=1}^s$, we can write $f_1$ and $f_2$ as follows via (7) in Lemma 2:

$$f_1(\mathbf{x}) = \sum_{i=1}^s g_i^{(1)}(\mathbf{x}_{\mathcal{I}_i}) + b^{(1)},$$

$$f_2(\mathbf{x}) = \sum_{i=1}^s g_i^{(2)}(\mathbf{x}_{\mathcal{I}_i}) + b^{(2)}.$$

Since $f = c_1 f_1 + c_2 f_2$,

$$
\begin{aligned}
f(\mathbf{x}) &= c_1 f_1(\mathbf{x}) + c_2 f_2(\mathbf{x}) \\
&= \left( \sum_{i=1}^s c_1 \times g_i^{(1)}(\mathbf{x}_{\mathcal{I}_i}) + c_1 \times b^{(1)} \right) + \left( \sum_{i=1}^s c_2 \times g_i^{(2)}(\mathbf{x}_{\mathcal{I}_i}) + c_2 \times b^{(2)} \right) \\
&= \sum_{i=1}^s \left( c_1 \times g_i^{(1)}(\mathbf{x}_{\mathcal{I}_i}) + c_2 \times g_i^{(2)}(\mathbf{x}_{\mathcal{I}_i}) \right) + c_1 b^{(1)} + c_2 b^{(2)}. 
\end{aligned}
\tag{13}
$$

By grouping terms as $g_i(\mathbf{x}_{\mathcal{I}_i}) = c_1 \times g_i^{(1)}(\mathbf{x}_{\mathcal{I}_i}) + c_2 \times g_i^{(2)}(\mathbf{x}_{\mathcal{I}_i})$ and $b = c_1 b^{(1)} + c_2 b^{(2)}$, we write (13) as

$$f(\mathbf{x}) = \sum_{i=1}^s g_i(\mathbf{x}_{\mathcal{I}_i}) + b. \tag{14}$$

From the form of (14), we can invoke (9): $\phi(\mathcal{I}_i) = g_i(\mathbf{x}^\star_{\mathcal{I}_i}) - g_i(\mathbf{x}'_{\mathcal{I}_i})$ via Lemma 2. This equation is rewritten as

$$
\begin{aligned}
\phi(\mathcal{I}_i) &= g_i(\mathbf{x}^\star_{\mathcal{I}_i}) - g_i(\mathbf{x}'_{\mathcal{I}_i}) \\
&= \left( c_1 \times g_i^{(1)}(\mathbf{x}^\star_{\mathcal{I}_i}) + c_2 \times g_i^{(2)}(\mathbf{x}^\star_{\mathcal{I}_i}) \right) - \left( c_1 \times g_i^{(1)}(\mathbf{x}'_{\mathcal{I}_i}) + c_2 \times g_i^{(2)}(\mathbf{x}'_{\mathcal{I}_i}) \right) \\
&= c_1 \left( g_i^{(1)}(\mathbf{x}^\star_{\mathcal{I}_i}) - g_i^{(1)}(\mathbf{x}'_{\mathcal{I}_i}) \right) + c_2 \left( g_i^{(2)}(\mathbf{x}^\star_{\mathcal{I}_i}) - g_i^{(2)}(\mathbf{x}'_{\mathcal{I}_i}) \right) \\
&= c_1 \phi_1(\mathcal{I}_i) + c_2 \phi_2(\mathcal{I}_i).
\end{aligned}
$$

By noting that $\mathcal{S} = \{\mathcal{I}_i\}_{i=1}^s$, this concludes the proof. $\qquad\square$

## F.4 Symmetry-Preserving

We first define *symmetric feature sets* as a generalization of "symmetric variables" from [47]. Feature index sets $\mathcal{I}_1$ and $\mathcal{I}_2$ are symmetric with respect to function $f$ if swapping features in $\mathcal{I}_1$ with the features in $\mathcal{I}_2$ does not change the function, This implies that for symmetric $\mathcal{I}_1$ and $\mathcal{I}_2$, their cardinalities are the same $|\mathcal{I}_1| = |\mathcal{I}_2|$, and they are disjoint sets in order to swap the features to any valid set index.

**Lemma 11** (Symmetry-Preserving). *For $\mathbf{x}^\star$ and $\mathbf{x}'$ that each have identical feature values between symmetric feature sets with respect to $f$, the symmetric feature sets receive identical attributions $\phi(\cdot)$.*

*Proof.* Since $\mathbf{x}^\star$ and $\mathbf{x}'$ each have identical feature values between the symmetric feature sets,

$$
\begin{aligned}
\{x_i^\star\}_{i \in \mathcal{I}_1} &= \{x_j^\star\}_{j \in \mathcal{I}_2}, \\
\{x_i'\}_{i \in \mathcal{I}_1} &= \{x_j'\}_{j \in \mathcal{I}_2}.
\end{aligned}
$$

Therefore, the symmetry implies the following for any $\mathbf{x}$ in the domain of $f$.

$$f\left(\mathbf{x}^\star_{\mathcal{I}_1} + \mathbf{x}'_{\mathcal{I}_2} + \mathbf{x}_{\backslash(\mathcal{I}_1 \cup \mathcal{I}_2)}\right) = f\left(\mathbf{x}'_{\mathcal{I}_1} + \mathbf{x}^\star_{\mathcal{I}_2} + \mathbf{x}_{\backslash(\mathcal{I}_1 \cup \mathcal{I}_2)}\right) \tag{15}$$

Setting $\mathbf{x} = \mathbf{x}'$, we rewrite (15) as

$$
\begin{aligned}
f\left(\mathbf{x}^\star_{\mathcal{I}_1} + \mathbf{x}'_{\mathcal{I}_2} + \mathbf{x}'_{\backslash(\mathcal{I}_1 \cup \mathcal{I}_2)}\right) &- f\left(\mathbf{x}'_{\mathcal{I}_1} + \mathbf{x}^\star_{\mathcal{I}_2} + \mathbf{x}'_{\backslash(\mathcal{I}_1 \cup \mathcal{I}_2)}\right) = 0 \\
&= f(\mathbf{x}^\star_{\mathcal{I}_1} + \mathbf{x}'_{\backslash\mathcal{I}_1}) - f(\mathbf{x}^\star_{\mathcal{I}_2} + \mathbf{x}'_{\backslash\mathcal{I}_2}) \\
&= \left(f(\mathbf{x}^\star_{\mathcal{I}_1} + \mathbf{x}'_{\backslash\mathcal{I}_1}) - f(\mathbf{x}')\right) - \left(f(\mathbf{x}^\star_{\mathcal{I}_2} + \mathbf{x}'_{\backslash\mathcal{I}_2}) - f(\mathbf{x}')\right) \\
&= \phi(\mathcal{I}_1) - \phi(\mathcal{I}_2)
\end{aligned}
$$

Therefore, $\phi(\mathcal{I}_1) = \phi(\mathcal{I}_2)$.

$\square$

# G  Discrete Mixed Partial Derivatives Detect Non-Additive Statistical Interactions

A generalized additive model $f_g$ is given by

$$f_g(\mathbf{x}) = \sum_{i=1}^{p} g_i(x_i) + b, \tag{16}$$

where $g_i(\cdot)$ can be any function of individual features $x_i$ and $b$ is a bias. Since each $x_i$ of $\mathbf{x} \in \mathcal{X}$ only takes on two values, a line can connect all valid points in each feature. Therefore, (16) is equivalent to

$$f_\ell(\mathbf{x}) = \sum_{i=1}^{p} w_i x_i + b, \tag{17}$$

for weights $w_i \in \mathbb{R}$ and the function domain being $\mathcal{X}$.

For the case where $p = 2$, the discrete mixed partial derivative is given by (3) or

$$\frac{\partial^2 f}{\partial x_1 \partial x_2} = \frac{1}{h_1 h_2}\left(f([x_1^\star, x_2^\star]) - f([x_1^\star, x_2']) - f([x_1', x_2^\star]) + f([x_1', x_2'])\right),$$

where $h_1 = |x_1^\star - x_1'|$ and $h_2 = |x_2^\star - x_2'|$. Since any three points (not on the same line) define a plane of the form (17) ($p = 2$), we can write the fourth point as having a function value with deviation $\delta$ from the plane.

$$
\begin{aligned}
\frac{\partial^2 f}{\partial x_1 \partial x_2} &= \frac{1}{h_1 h_2}\left(f([x_1^\star, x_2^\star]) - f([x_1^\star, x_2']) - f([x_1', x_2^\star]) + f([x_1', x_2'])\right) \\
&= \frac{1}{h_1 h_2}\left((w_1 x_1^\star + w_2 x_2^\star + b + \delta) - (w_1 x_1^\star + w_2 x_2' + b) - (w_1 x_1' + w_2 x_2^\star + b)\right. \tag{18} \\
&\qquad\qquad \left. + (w_1 x_1' + w_2 x_2' + b)\right) \\
&= \frac{\delta}{h_1 h_2}.
\end{aligned}
$$

If (18) is 0, then $\delta = 0$, which implies that $f$ can be written as (17). $\delta \neq 0$ implies the opposite, that $f$ cannot be written in linear form (by definition). Since (17) is equivalent to (16) in the domain of $\mathcal{X}$, this implies that $\delta \neq 0$ if and only if $f(\mathbf{x}) \neq g_1(x_1) + g_2(x_2) + b$.

Based on Def. 1, we can conclude that a nonzero discrete mixed partial derivative w.r.t. $x_1$ and $x_2$ in the space $\mathcal{X}$ at $p = 2$ detects a non-additive statistical interaction between the two features.

For the case where $p > 2$, Def. 1 states that a pairwise interaction $\{i, j\}$ exists in $f$ if and only if $f(\mathbf{x}) \neq f_i(\mathbf{x}_{\backslash\{i\}}) + f_j(\mathbf{x}_{\backslash\{j\}})$ for functions $f_i(\cdot)$ and $f_j(\cdot)$. This means that $\{i, j\}$ is declared to be an interaction if a local $\{i, j\}$ interaction occurs at any $\mathbf{x}_{\backslash\{i,j\}}, \mathbf{x} \in \mathcal{X}$.

Therefore, we can detect non-additive statistical interactions $\{i, j\}$ for general $p \geq 2$ via

$$\mathbb{E}_{\mathbf{x}} \left[ \frac{\partial^2 f}{\partial x_i \partial x_j} \right]^2 > 0,$$

which mirrors the definition of pairwise interaction for real-valued $\mathbf{x}$ in [18].

## H   Early Works on Feature Interaction Interpretation

We discuss early works on feature interaction interpretation and provide a timeline for this research history in Table 4. We also discuss mixed partial derivatives on dichotomous variables in H.3.

### H.1   Origins

The notion of a feature interaction has been studied at least since the 19th century when John Lawes and Joseph Gilbert used factorial designs in agricultural research at the Rothamsted Experimental Station [11]. A factorial design is an experiment that includes observations at all combinations of categories of each factor or feature. However, the "advantages [of factorial design] had never been clearly recognised, and many research workers believed that the best course was the conceptually simple one of investigating one question at a time" [58]. In the early 20th century, Fisher et al. (1926) [17] emphasized the importance of factorial designs as being the only way to obtain information about feature interactions. Near the same time, Fisher (1921) [15] also developed one of the foundations of statistical analysis called Analysis of Variance (ANOVA) including two-way ANOVA [16], which is a factorial method to detect pairwise feature interactions based on differences among group means in a dataset. Tukey (1949) [52] extended two-way ANOVA to test if two categorical features are non-additively related to the expected value of a outcome variable. This work set a precedent for later research on detecting feature interactions based on their non-additive definition. Soon after, experimental designs were generalized to study feature interactions, in particular the generalized randomized block design [55], which assigns test subjects to different categories (or blocks) between features in a way where cross-categories between features serve as interaction terms in linear regression.

There was a surge of interest in improving the analysis of feature interactions after the mid 20th century. Belsion (1959) [5] and Morgan & Sonquist (1963) [35] proposed Automatic Interaction Detection (AID) originally under a different name. AID detects interactions by subdividing data into disjoint exhaustive subsets to model an outcome based on categorical features. Based on AID, Kass (1980) [27] developed Chi-square Automatic Interaction Detection (CHAID), which determines how categorical features best combine in decision trees via a chi-square test. AID and CHAID were precursors to modern decision tree prediction models. Concurrently, Nelder (1977) [37] introduced the "Principle of Marginality" arguing that a feature interaction and its marginal variables should not be considered separately, for example in linear regression. Hamada & Wu (1992) [22] provided a contrasting view that an interaction is only important if one or both of its marginal variables are important. Around the same time, an influential book on interpreting feature interactions was published on how to test, plot, and understand interactions of two or three continuous or categorical features [3].

### H.2   Early 21st Century Works

At the start of the 21st century, efforts began to focus on interpreting interactions in accurate prediction models. Ai & Norton (2003) [2] proposed extracting interactions from logit and probit models via mixed partial derivatives. Gevrey (2006) [19] followed up by proposing mixed partial derivatives to extract interactions from multilayer perceptrons with sigmoid activations when at the time, only shallow neural networks were studied. Friedman & Popescu (2008) [18] proposed using hybrid models to capture interactions with decision trees and univariate effects with linear regression. Sorokina et al. (2008) [46] proposed to use high-performance additive trees to detect feature interactions based on their non-additive definition. At the turn of the decade, we saw Bien et al. [6] capture interactions with different heredity conditions using a hierarchical lasso on linear regression models. Then, Hao & Zhang (2014) [23] drew attention towards interaction screening in high dimensional data. This summarizes feature interaction research before 2015.

### H.3 Note on Mixed Partial Derivatives on Dichotomous Variables

To our knowledge, the usage of mixed partial derivatives for interaction detection on dichotomous variables (features that only take two possible values) originated at the turn of the 21st century [2, 20], but existing methods rely on single contexts [2] or random contexts [14, 20]. Furthermore, these methods do not consider the union of overlapping pairwise interactions for disjoint higher-order interaction detection. Our choice of contexts and our disjoint interaction detection are both important to the `Archipelago` framework, as we discussed in §4.2 and showed through axiomatic analysis (§3.2) and experiments (§5.2).

TABLE 4    Timeline of research on feature interaction interpretation (Pre-2015)

| | | |
|---|---|---|
| *Lawes & Gilbert* - factorial design in agricultural research at the Rothamsted Experimental Station | 1843 | |
| *Fisher* - two-way Analysis of Variance (ANOVA) | 1925 | |
| | 1949 | *Tukey* - Tukey's test of additivity |
| | 1955 | *Wilk* - generalized random block design |
| *Belson* - Automatic Interaction Detection by subdividing data | 1959 | |
| *Nelder* - Principle of Marginality | 1977 | |
| | 1980 | *Kass* - Chi-square Automatic Interaction Detection by combining features in decision trees via chi-square tests |
| | 1991 | *Aiken & West* - book on interpreting interaction effects |
| *Hamada & Wu* - heredity conditions | 1992 | |
| *Ai & Norton* - interactions in logit and probit models | 2003 | |
| | 2006 | *Gevry et al.* - interactions in sigmoid neural networks |
| *Friedman & Popescu* - RuleFit to detect interactions by mixing linear regression and trees | 2008 | *Sorokina et al.* - Additive Groves to detect non-additive interactions |
| *Bien et al.* - Hierarchical Lasso | 2013 | |
| *Hao & Zhang* - interaction screening in high dimensional data | 2014 | |

# I    Attributions Compared to Annotation Labels

(a) Word $\rho$

(b) Phrase $\rho$

Figure 9: Text explanation metrics ((a) Word $\rho$ and (b) Phrase $\rho$) versus top and bottom % of attributions retained for different attribution methods on BERT over the SST test set. These plots expand the analysis of Table 2.

Figure 10: Image explanation metric (segment AUC) versus top and bottom % of attributions retained for different attribution methods on ResNet152 over the MS COCO test set. These plots expand the analysis of Table 2.

# J    Visualization Comparisons

## J.1    Sentiment Analysis

Visualization comparisons of different attribution methods on BERT are shown in Figs. 12-16 for random test sentences from SST. The visualization format is the same as Fig. 4. Note that all *individual* feature attributions that correspond to stop words (from [33]) are omitted in these comparisons and Figs. 1, 4.

## J.2    Image Classification

In Fig. 11, we visualize `Archipelago` explanations on $\mathcal{S}$ via top-5 pairwise interactions (§4.2.2), where positive attribution interactions are shown for clarity. The images are randomly selected from the ImageNet test set. It is interesting to see which image parts interact, such as the eyes of the "great dane" image.

Visualization comparisons of different attribution methods on ResNet152 are shown in Figs. 17-21 for the same random test images from ImageNet.

# K    `ArchDetect` Ablation Visualizations

We run an ablation study removing the $\mathbf{x}'_{\backslash\{i,j\}}$ baseline context from (5) for disjoint interaction detection and examine its effect on visualizations. The visualizations are shown in Fig. 22 for

Figure 11: Our ResNet152 visualizations on random test images from ImageNet. Colored outlines indicate interactions with positive attribution. $f_c$ is the image classification result. To our knowledge, only this work shows interactions that support the image classification via interaction attribution.

sentiment analysis and Figs. 23 and 24 for image classification. Top-3 and top-5 pairwise interactions are used in sentiment analysis and image classification respectively before merging the interactions.

## L `ArchAttribute` with Different Interaction Detectors

We compare visualizations of `ArchAttribute` using different interaction detectors. The visualizations are shown in Figs. 25 and 26 for sentiment analysis and Figs. 27, 28, and 29 for image classification. Top-3 and top-5 pairwise interactions are used in sentiment analysis and image classification respectively before merging the interactions.

Text input: "I regret to report that these ops are just not extreme enough ."     Classification: neg

Figure 12: Text Viz. Comparison A. In the first text example, "regret, not extreme enough" is a meaningful and strongly negative interaction. In the second example, "when you begin to" interacts to diminish its overall attribution magnitude.

Text input: "It 's solid and affecting and exactly as thought-provoking as it should be ."     Classification: pos

Archipelago   it ' s solid and affecting and exactly as thought - pro -voking as it should be .

Difference +
ArchDetect    it ' s solid and affecting and exactly as thought - pro -voking as it should be .

IG            it ' s solid and affecting and exactly as thought - pro -voking as it should be .

IG +
ArchDetect    it ' s solid and affecting and exactly as thought - pro -voking as it should be .

IH

LIME          it ' s solid and affecting and exactly as thought - pro -voking as it should be .

MAHE

SI            it ' s solid and affecting and exactly as thought - pro -voking as it should be .

STI           it ' s solid and affecting and exactly as thought - pro -voking as it should be .

Text input: "A lousy movie that 's not merely unwatchable , but also unlistenable ."     Classification: neg

Archipelago   a lou -sy movie that ' s not merely un -watch -able , but also un -list -ena -ble .

Difference +
ArchDetect    a lou -sy movie that ' s not merely un -watch -able , but also un -list -ena -ble .

IG            a lou -sy movie that ' s not merely un -watch -able , but also un -list -ena -ble .

IG +
ArchDetect    a lou -sy movie that ' s not merely un -watch -able , but also un -list -ena -ble .

IH

LIME          a lou -sy movie that ' s not merely un -watch -able , but also un -list -ena -ble .

MAHE

SI            a lou -sy movie that ' s not merely un -watch -able , but also un -list -ena -ble .

STI           a lou -sy movie that ' s not merely un -watch -able , but also un -list -ena -ble .

Figure 13: Text Viz. Comparison B. In the first text example, "thought provoking" is a meaningful and strongly positive interaction. In the second example, the "lousy, un" interaction factors in a large context to make a negative text classification.

Text input: "Tsai Ming-liang has taken his trademark style and refined it to a crystalline point ."     Classification: pos

Figure 14: Text Viz. Comparison C. In the first text example, "refined, to a crystalline" is a meaningful and strongly positive interaction. In the second example, "is aptly named" is also a meaningful and strongly positive interaction.

Text input: "The ending is a cop-out ."     Classification: neg

Archipelago        the ending is a cop - out .

Difference +
ArchDetect         the ending is a cop - out .

IG                 the ending is a cop - out .

IG +
ArchDetect         the ending is a cop - out .

IH

LIME               the ending is a cop - out .

MAHE

SI                 the ending is a cop - out .

STI                the ending is a cop - out .

Text input: "A feel-good picture in the best sense of the term ."     Classification: pos

Archipelago        a feel - good picture in the best sense of the term .

Difference +
ArchDetect         a feel - good picture in the best sense of the term .

IG                 a feel - good picture in the best sense of the term .

IG +
ArchDetect         a feel - good picture in the best sense of the term .

IH

LIME               a feel - good picture in the best sense of the term .

MAHE

SI                 a feel - good picture in the best sense of the term .

STI                a feel - good picture in the best sense of the term .

Figure 15: Text Viz. Comparison D. In the first text example, "the ending, out" is a meaningful and negative interaction. In the second example, "a feel good, best" is a meaningful and strongly positive interaction.

Text input: "All prints of this film should be sent to and buried on Pluto ."      Classification: neg

Figure 16: Text Viz. Comparison E. In the first text example, "film should be, buried" is a meaningful and strongly negative interaction. In the second example, "-oherent" belongs to a negative word "incoherent".

Figure 17: Image Viz. Comparison A. In the first image example, the dog's eyes are a meaningful interaction supporting the classification. In the second example, the monkey's head is also a positive interaction.

Figure 18: Image Viz. Comparison B. In the first image example, the obelisk tip is a meaningful interaction supporting the classification. In the second example, the leopard's face is also a positive interaction.

Figure 19: Image Viz. Comparison C. In the first image example, different patches of the apron are interactions supporting the classification. In the second example, the stork's body is an interaction that strongly supports the classification.

Image input

Classification: waffle iron

for classification

against classification

Archipelago
individual effects | interaction $\mathcal{I}_1$ | interaction $\mathcal{I}_2$

Difference + ArchDetect
individual effects | interaction $\mathcal{I}_1$ | interaction $\mathcal{I}_2$

IG + ArchDetect
individual effects ($\times 10$) | interaction $\mathcal{I}_1$ | interaction $\mathcal{I}_2$

LIME
individual effects

MAHE
interaction $\mathcal{I}_1$ | interaction $\mathcal{I}_2$ | interaction $\mathcal{I}_3$

SI
individual effects | interaction $\mathcal{I}_1$ | interaction $\mathcal{I}_2$ | interaction $\mathcal{I}_3$ | interaction $\mathcal{I}_4$ | interaction $\mathcal{I}_5$

STI
individual effects | interaction $\mathcal{I}_1$ | interaction $\mathcal{I}_2$ | interaction $\mathcal{I}_3$ | interaction $\mathcal{I}_4$ | interaction $\mathcal{I}_5$

Image input

Classification: snow leopard, ounce, Panthera uncia

Archipelago
individual effects | interaction $\mathcal{I}_1$ | interaction $\mathcal{I}_2$

Difference + ArchDetect
individual effects | interaction $\mathcal{I}_1$ | interaction $\mathcal{I}_2$

IG + ArchDetect
individual effects ($\times 10$) | interaction $\mathcal{I}_1$ | interaction $\mathcal{I}_2$

LIME
individual effects

MAHE
interaction $\mathcal{I}_1$ | interaction $\mathcal{I}_2$ | interaction $\mathcal{I}_3$

SI
individual effects | interaction $\mathcal{I}_1$ | interaction $\mathcal{I}_2$ | interaction $\mathcal{I}_3$ | interaction $\mathcal{I}_4$ | interaction $\mathcal{I}_5$

STI
individual effects | interaction $\mathcal{I}_1$ | interaction $\mathcal{I}_2$ | interaction $\mathcal{I}_3$ | interaction $\mathcal{I}_4$ | interaction $\mathcal{I}_5$

Figure 20: Image Viz. Comparison D. In the first image example, certain small patches of the waffle iron interact, one of which supports the classification. In the second example, the leopard's face is the primary positive interaction.

Figure 21: Image Viz. Comparison E. In the first image example, different parts of the polaroid camera are interactions that positively support the classification. In the second example, the dogs' heads and body are also positive interactions.

Text input: "I regret to report that these ops are just not extreme enough ."     Classification: neg

w/ Baseline Context
i regret to report that these ops are just not extreme enough .

w/o Baseline Context
i regret to report that these ops are just not extreme enough .

pos
neg

Text input: "It 's a worse sign when you begin to envy her condition ."     Classification: neg

w/ Baseline Context
it ' s a worse sign when you begin to envy her condition .

w/o Baseline Context
it ' s a worse sign when you begin to envy her condition .

Text input: "It 's solid and affecting and exactly as thought-provoking as it should be ."     Classification: pos

w/ Baseline Context
it ' s solid and affecting and exactly as thought - pro -voking as it should be .

w/o Baseline Context
it ' s solid and affecting and exactly as thought - pro -voking as it should be .

Text input: "A lousy movie that 's not merely unwatchable , but also unlistenable ."     Classification: neg

w/ Baseline Context
a lou -sy movie that ' s not merely un -watch -able , but also un -list -ena -ble .

w/o Baseline Context
a lou -sy movie that ' s not merely un -watch -able , but also un -list -ena -ble .

Text input: "Tsai Ming-liang has taken his trademark style and refined it to a crystalline point ."     Classification: pos

w/ Baseline Context
ts -ai ming - liang has taken his trademark style and refined it to a crystalline point .

w/o Baseline Context
ts -ai ming - liang has taken his trademark style and refined it to a crystalline point .

Text input: "As an actor , The Rock is aptly named ."     Classification: pos

w/ Baseline Context
as an actor , the rock is apt -ly named .

w/o Baseline Context
as an actor , the rock is apt -ly named .

Text input: "The ending is a cop-out ."     Classification: neg

w/ Baseline Context
the ending is a cop - out .

w/o Baseline Context
the ending is a cop - out .

Text input: "A feel-good picture in the best sense of the term ."     Classification: pos

w/ Baseline Context
a feel - good picture in the best sense of the term .

w/o Baseline Context
a feel - good picture in the best sense of the term .

Text input: "All prints of this film should be sent to and buried on Pluto ."     Classification: neg

w/ Baseline Context
all prints of this film should be sent to and buried on pluto .

w/o Baseline Context
all prints of this film should be sent to and buried on pluto .

Figure 22: Text Viz. with `ArchDetect` Ablation. The interactions tend to use more salient words when including the baseline context, which is proposed in `ArchDetect`.

Figure 23: Image Viz. with `ArchDetect` Ablation A. The interactions tend to focus more on salient patches of the images when including the baseline context, which is proposed in `ArchDetect`.

Figure 24: Image Viz. with `ArchDetect` Ablation B. The interactions tend to focus on salient patches of the images when including the baseline context.

Text input: "I regret to report that these ops are just not extreme enough ."    Classification: neg

ArchDetect    i regret to report that these ops are just not extreme enough .

IH    i regret to report that these ops are just not extreme enough .

MAHE    i regret to report that these ops are just not extreme enough .

SI    i regret to report that these ops are just not extreme enough .

STI    i regret to report that these ops are just not extreme enough .

Text input: "It 's a worse sign when you begin to envy her condition ."    Classification: neg

ArchDetect    it ' s a worse sign when you begin to envy her condition .

IH    it ' s a worse sign when you begin to envy her condition .

MAHE    it ' s a worse sign when you begin to envy her condition .

SI    it ' s a worse sign when you begin to envy her condition .

STI    it ' s a worse sign when you begin to envy her condition .

Text input: "It 's solid and affecting and exactly as thought-provoking as it should be ."    Classification: pos

ArchDetect    it ' s solid and affecting and exactly as thought - pro -voking as it should be .

IH    it ' s solid and affecting and exactly as thought - pro -voking as it should be .

MAHE    it ' s solid and affecting and exactly as thought - pro -voking as it should be .

SI    it ' s solid and affecting and exactly as thought - pro -voking as it should be .

STI    it ' s solid and affecting and exactly as thought - pro -voking as it should be .

Text input: "A lousy movie that 's not merely unwatchable , but also unlistenable ."    Classification: neg

ArchDetect    a lou -sy movie that ' s not merely un -watch -able , but also un -list -ena -ble .

IH    a lou -sy movie that ' s not merely un -watch -able , but also un -list -ena -ble .

MAHE    a lou -sy movie that ' s not merely un -watch -able , but also un -list -ena -ble .

SI    a lou -sy movie that ' s not merely un -watch -able , but also un -list -ena -ble .

STI    a lou -sy movie that ' s not merely un -watch -able , but also un -list -ena -ble .

Text input: "Tsai Ming-liang has taken his trademark style and refined it to a crystalline point ."    Classification: pos

ArchDetect    ts -ai ming - liang has taken his trademark style and refined it to a crystalline point .

IH    ts -ai ming - liang has taken his trademark style and refined it to a crystalline point .

MAHE    ts -ai ming - liang has taken his trademark style and refined it to a crystalline point .

SI    ts -ai ming - liang has taken his trademark style and refined it to a crystalline point .

STI    ts -ai ming - liang has taken his trademark style and refined it to a crystalline point .

Figure 25: Text Viz. with `ArchAttribute` + Different Interaction Detectors A.

Text input: "As an actor , The Rock is aptly named ."     Classification: pos

Text input: "The ending is a cop-out ."     Classification: neg

Text input: "A feel-good picture in the best sense of the term ."     Classification: pos

Text input: "All prints of this film should be sent to and buried on Pluto ."     Classification: neg

Text input: "Arguably the year 's silliest and most incoherent movie ."     Classification: neg

Figure 26: Text Viz. with `ArchAttribute` + Different Interaction Detectors B.

Image input

Classification: Great Dane

for classification
against classification

ArchDetect
individual effects    interaction $\mathcal{I}_1$

MAHE
individual effects    interaction $\mathcal{I}_1$

SI
individual effects    interaction $\mathcal{I}_1$    interaction $\mathcal{I}_2$    interaction $\mathcal{I}_3$    interaction $\mathcal{I}_4$

STI
individual effects    interaction $\mathcal{I}_1$    interaction $\mathcal{I}_2$    interaction $\mathcal{I}_3$    interaction $\mathcal{I}_4$

Image input

Classification: spider monkey, Ateles geoffroyi

ArchDetect
individual effects    interaction $\mathcal{I}_1$    interaction $\mathcal{I}_2$

MAHE
individual effects    interaction $\mathcal{I}_1$

SI
individual effects    interaction $\mathcal{I}_1$    interaction $\mathcal{I}_2$    interaction $\mathcal{I}_3$    interaction $\mathcal{I}_4$    interaction $\mathcal{I}_5$

STI
individual effects    interaction $\mathcal{I}_1$    interaction $\mathcal{I}_2$    interaction $\mathcal{I}_3$

Image input

Classification: obelisk

ArchDetect
individual effects    interaction $\mathcal{I}_1$    interaction $\mathcal{I}_2$

MAHE
individual effects    interaction $\mathcal{I}_1$

SI
individual effects    interaction $\mathcal{I}_1$    interaction $\mathcal{I}_2$    interaction $\mathcal{I}_3$    interaction $\mathcal{I}_4$    interaction $\mathcal{I}_5$

STI
individual effects    interaction $\mathcal{I}_1$    interaction $\mathcal{I}_2$    interaction $\mathcal{I}_3$    interaction $\mathcal{I}_4$

Image input

Classification: snow leopard, ounce, Panthera uncia

ArchDetect
individual effects    interaction $\mathcal{I}_1$    interaction $\mathcal{I}_2$    interaction $\mathcal{I}_3$

MAHE
individual effects    interaction $\mathcal{I}_1$

SI
individual effects    interaction $\mathcal{I}_1$    interaction $\mathcal{I}_2$    interaction $\mathcal{I}_3$    interaction $\mathcal{I}_4$    interaction $\mathcal{I}_5$

STI
individual effects    interaction $\mathcal{I}_1$    interaction $\mathcal{I}_2$    interaction $\mathcal{I}_3$

Figure 27: Image Viz. with `ArchAttribute` + Different Interaction Detectors A.

Image input

Classification: apron

for classification

against classification

ArchDetect
individual effects    interaction $\mathcal{I}_1$    interaction $\mathcal{I}_2$

MAHE
individual effects    interaction $\mathcal{I}_1$

SI
individual effects    interaction $\mathcal{I}_1$    interaction $\mathcal{I}_2$    interaction $\mathcal{I}_3$    interaction $\mathcal{I}_4$

STI
individual effects    interaction $\mathcal{I}_1$    interaction $\mathcal{I}_2$    interaction $\mathcal{I}_3$    interaction $\mathcal{I}_4$

Image input

Classification: black stork, Ciconia nigra

ArchDetect
individual effects    interaction $\mathcal{I}_1$

MAHE
individual effects    interaction $\mathcal{I}_1$

SI
individual effects    interaction $\mathcal{I}_1$    interaction $\mathcal{I}_2$    interaction $\mathcal{I}_3$    interaction $\mathcal{I}_4$

STI
individual effects    interaction $\mathcal{I}_1$

Image input

Classification: waffle iron

ArchDetect
individual effects    interaction $\mathcal{I}_1$    interaction $\mathcal{I}_2$

MAHE
individual effects    interaction $\mathcal{I}_1$

SI
individual effects    interaction $\mathcal{I}_1$    interaction $\mathcal{I}_2$    interaction $\mathcal{I}_3$    interaction $\mathcal{I}_4$    interaction $\mathcal{I}_5$

STI
individual effects    interaction $\mathcal{I}_1$    interaction $\mathcal{I}_2$    interaction $\mathcal{I}_3$    interaction $\mathcal{I}_4$

Image input

Classification: snow leopard, ounce, Panthera uncia

ArchDetect
individual effects    interaction $\mathcal{I}_1$    interaction $\mathcal{I}_2$

MAHE
individual effects    interaction $\mathcal{I}_1$

SI
individual effects    interaction $\mathcal{I}_1$    interaction $\mathcal{I}_2$    interaction $\mathcal{I}_3$    interaction $\mathcal{I}_4$    interaction $\mathcal{I}_5$

STI
individual effects    interaction $\mathcal{I}_1$    interaction $\mathcal{I}_2$    interaction $\mathcal{I}_3$

Figure 28: Image Viz. with `ArchAttribute` + Different Interaction Detectors B.

Figure 29: Image Viz. with `ArchAttribute` + Different Interaction Detectors C.