[Reviews · NeurIPS 2020]

Review 1

Summary and Contributions: This paper provides a method to generate interpretable attribution for higher order interactions and as well as discovering these interactions in an efficient manner. Main contributions are : 1. Attribution method : This is simply defined as f(selected features + baseline) - f(baseline) . The authors provide certain proofs of axioms for attribution - completeness , set attribution, etc. 2. Detection: This method based on mixed partial derivatives is responsible for detecting higher order interactions (i.e are there any separable variables within a given feature set). 3. Run experiments on constructed datasets and two existing datasets - (on sentiment analysis and image classification).

Strengths: 1. Both methods for attribution and detection are novel and I cannot find any specific problem in the proofs. While I am not sure about why the feature attribution method is superior in theoretical terms, The empirical findings do indicate improvement of existing methods . 2. Novelty: The detection method is novel in terms of scalability, although assumption 5 is suspect on which the method rest.

Weaknesses: 1. Relevance of the theorems: While I find principled approach to I&E methods useful, I fail to see the relevance of the axioms that the authors prove. It would be better of the authors had provided examples (even theoretical) when such considerations fail to provide correct intuition of the model (or atleast reduce the trust in the model). Update: Thanks for the clarification. While we intuitively may want some of these properties, neither this paper nor the related work the authors pointed to explicitly test for what goes wrong when these properties are invalidated (other than hand wavy heatmap based explanations). But this is a problem bigger than this paper in interpretability field. Adding a discussion or examples where not having these properties show breakdown of coherence would make this paper stronger. 2. Assumption 5: Consider the case "not a very good movie". Set {_, a, very, good, _ } and set {not, _, very, _, _} both have interaction effects. Can you clarify if such contexts are captured by context of x' or x* ? Thanks for the clarification

Correctness: See above. I don't have any specific concerns about the empirical methodology. The author's use standard datasets for evaluation and against multiple existing baselines.

Clarity: I didn't find any clarity issues.

Relation to Prior Work: The authors have cited, to the best of my knowledge, all relevant work and positioned there paper in context of previous work.

Reproducibility: Yes

Additional Feedback: I am not sure how much do the method depend on the choice of baseline. Do you see the same interaction effects if say you move from all black to all white baseline ? Would the conclusions drastically change ? Can you clarify how assumption 5 ties to detecting higher order interaction ? What would the big-O runtime for nth order interaction detection ?


Review 2

Summary and Contributions: The paper proposes a feature interaction detection and attribution framework called Archipelago. It comprises of two components: the interaction attribution method called ArchAttribute and the interaction detector called ArchDetect. The framework is model-agnostic, interpretable, and obeys standard attribution axioms.

Strengths: 1. The interaction detector method, ArchDetector, is scalable to higher-order feature interactions, where the number of features is greater than two. It does so by merging pairwise interactions for disjoint arbitrary order detection. 2. The attribution method, ArchAttribute, proposes a new attribution measure for feature interaction. The measure satisfies attribution axioms and is interpretable. The paper also proposes an additional axiom called Set Attribution and extends the axioms proposed for individual features to feature sets. 3. The authors compare their proposed technique with prior interaction attribution methods and use text and image based models for empirical validation of their work. A rich set of ablation results is provided in the appendix. 4. Appendix H is interesting and nicely summarizes the work that has happened in this field.

Weaknesses: 1. I found the main paper difficult to comprehend and this is after multiple readings. The key ideas should be made more accessible. 2. Subtracting out the effect of the baseline features makes attribution assignment more precise but it is not clear why ArchAttribute (Eq. 2) should also be more interpretable? 3. How does ArchDetect (in particular the technique mentioned in 4.2.2) detect arbitrary size disjoint feature sets for images? In Sec 5.3 (Image Classification), image segments seem to be the feature sets. More clarity on this would be helpful.

Correctness: Yes

Clarity: The paper can be written better.

Relation to Prior Work: Yes

Reproducibility: Yes

Additional Feedback: ++ post rebuttal comments I have read the rebuttal and would like to thank the authors for their responses. I will keep my score unchanged.


Review 3

Summary and Contributions: This paper proposes Archipelago, a new feature interaction detection and attribution framework for explaining predictions of machine learning models. Basically, Archipelago consists of two main components. First, ArchAttribute calculates an attribution score for a given set of feature indices in a way that satisfies attribution axioms. Second, ArchDetect efficiently detects interactions between features and constructs disjoint interaction sets. The experiments were conducted on sentiment analysis and image classification tasks. Additional qualitative results on COVID-19 classification and ad-recommendation were also discussed in the paper.

Strengths: - The proposed framework satisfies attribution axioms, while being model-agnostic and efficient to compute. - In addition to synthetic validation for ArchDetect, the experiments were also conducted on several real-world datasets and compared the proposed framework with many related work.

Weaknesses: To compute some measures in Table 2, the authors compared interaction attributions with ground truth labels (SST phrase labels, image segment labels). This may not be appropriate because it could be the case that the model comes to the prediction using reasons which are different from humans (i.e., the ground truths). Therefore, a low score does not always mean that the attribution method is not faithful. Similarly, for word correlation, I have no idea why we can use coefficients of a linear model as a reference to compare attribution scores.

Correctness: I'm not familiar with the work in this area, so I cannot firmly verify the correctness of this paper.

Clarity: The paper is quite difficult to read. Personally, I believe that more background and examples would be helpful.

Relation to Prior Work: The paper distinguishes itself from prior work in many aspects including being interpretable, model-agnostic, axiomatic, and efficient. Overall, the related works section provides a concise discussion and sufficient details can be found in the appendix.

Reproducibility: Yes

Additional Feedback: - The authors claim that the proposed method is more interpretable and axiomatic. I would like to suggest defining "interpretability" of explanation method in the introduction because it may not be well understood in general, in contrast to interpretability of prediction models. Besides, I understand that "axiomatic" here means "satisfying the axioms". However, other methods also satisfy some (but not all the) axioms discussed in this paper and the appendix. So, it would be better to clearly indicate/discuss the important axioms which are satisfied by the proposed method in the introduction. - As efficiency is one of the strengths of the proposed framework, runtime comparison should also be discussed, at least briefly, in the main paper (in addition to the details in the appendix). - Regarding the two functions f at line 145 and 146, why are they functionally equivalent considering that f1 and f2 can be a lot more complex than a coefficient times a variable? - Section 4.2.1 explains how to estimate the interaction strength between features i and j. How large is the strength score needed to conclude that feature i and j have an interaction? Is there any threshold? After reading the author response: Thank you for addressing our concerns. Overall, I decided to keep my score the same. If the paper is accepted, I hope the authors provide more background and examples in the next version (e.g., by adding some important answers, like Why these axioms?, from the author response to the paper).


Review 4

Summary and Contributions: The paper proposes an interaction attribution and detection framework called Archipelago. Specifically, the paper proposes a feature attribution measure called (ArchAttribute), a new axiom, and a feature interaction detector (ArchDetect). The framework has been shown to have better performance than the state-of-the-art [Table 2] in sentiment analysis (using BERT on SST) and image classification (using ResNet on ImageNet). ArchDetect achieves better performance at interaction detection than state-of-the-art [Table 1]. The paper also presents a couple of examples of feature attributions (e.g. Figures 4-6). The claims that the proposed approach is significantly more interpretable but there is no clear evidence of this.

Strengths: In not an expert in the area of interaction detection, but I found the paper and the ideas reasonably easy to follow. The links to existing works on interaction detection are clear, and the ideas are coherent. The paper does an excellent job of showing that in the space of interaction detection, the presented framework achieves better results. From the discussion and results, I am convinced that ArchAttribute and ArchDetect are novel.

Weaknesses: Two key problems lead me to make my final decision. First, I did not see a clear link between "...attribution scores to ground truth annotation labels on subsets of features..." and interpretability. The paper starts by claiming that the proposed framework is interpretable. However, the only link that I found to interpretability was that above statement. To me, it was unclear as to why the measures used in the paper lead to more interpretability. For me, there is no evidence to convince me that the framework enhances interpretability. Second, and leading from the first point, both the title and the framing is misleading. For me, the paper was primarily on pairwise interaction attribution and detection (which I feel that paper did an excellent job at), but the framing of the paper was around interpretability, and as I pointed to before, in my view, there is no evidence of that. As a minor remark, the paper presents some complexity analysis starting on line 179. Would it possible for the authors to include results of actual experiments comparing the performance of their method with the state-of-the-art? There may be a few typos, e.g. interpetability (line 222).

Correctness: As pointed to earlier, work lacks empirical evaluation to demonstrate that the framework improves interpretability. Without clear evidence that measures used in the paper link to interpretability, it is challenging to verify the claim. May be a human study is worth considering?

Clarity: The clarity of the paper is excellent. I found the paper easy to read and understand.

Relation to Prior Work: There are clear links to prior work and state-of-the-art methods as far as interaction detection is concerned.

Reproducibility: Yes

Additional Feedback: I thank the authors for their rebuttal. As the authors stated, they will include the discussion on the link between interpretability and ground truth annotations in the revised version. I am satisfied with this response and I have reflected this in my score.


Review 5

Summary and Contributions: This paper introduces methods for identifying sets of feature interactions which are important to a prediction, and axiomatically attributing prediction to those sets.

Strengths: Theoretical grounding strengths: - it generalizes single-feature attribution axioms to the case of interactions, and verifies its proposed method satisfies them - although I'm a bit skeptical of the assumption of the existence of a meaningful "baseline" input whose features don't interact with the input being explained, at least they explicitly state their assumptions. Empirical evaluation strengths: - it validates the method on synthetic examples with ground truth - its real-world evaluation metrics (e.g. in Table 2) are clever and compelling - it tests against a wide variety of baselines - it tests on a wide variety of datasets - it shows many examples that help us qualitatively understand the differences between the methods Relevance: - the paper addresses a relevant problem in the literature on explaining nonlinear model predictions. Most techniques explain such predictions in terms of individual features, or in certain cases, pairwise interactions, despite the fact that we use nonlinear models because of their ability to take advantage of more complicated interactions. It makes a lot of sense to have an explanation method which can identify such interactions when appropriate. Significance and novelty: - The whole area of interaction attribution is still fairly new, especially wrt. interactions higher-order than pairwise.

Weaknesses: Theoretical grounding: - I'm not confident Assumption 5 is always satisfied; I think for DNNs on common datasets, it's nontrivial to find a baseline that is actually noninformative / unimportant. Would it be possible to test how well this is satisfied? - In the experiments, it's necessary to choose an interaction strength _threshold_ for combining pairwise interactions (e.g. top and bottom 10%). This choice seems critical to any user-facing visualization, especially since realistic models will probably have nonzero interaction strengths for all pairs. Although there are some results in Appendix I showing how quantitative results vary with the threshold, the paper does not discuss how to meaningfully choose this value. - I'm not 100% convinced that meaningful higher-order interactions will always be detected by merging pairwise interactions (though perhaps this is covered by Assumption 5). As a concrete example, imagine a function in R^40 like in the synthetic examples where x* is [1,1,...,1], x' is [-1,-1,...,-1], and there is a component in the function that takes a subset of dimensions (say the first 10) and returns 1 if and only if exactly half of those 10 dimensions have positive values. In that case, there is a meaningful interaction between those 10 dimensions, but I don't think it would be returned by the method. It would be great if you could either explain how I'm incorrect or how this example violates the method's explicitly stated assumptions. Empirical evaluation: - The paper introduces two new methods: ArchDetect (for finding sets of interacting features), and ArchAttribute (for computing feature set attributions), which together seem to result in compelling performance. Ironically, though, the paper doesn't make it clear whether this performance improvement is primarily because of ArchDetect, ArchAttribute, or their interaction :) - To fix this, I would strongly recommend showing results for ArchAttribute with a detection method other than ArchDetect (e.g. using Hessian components or Shapley interaction indices as pairwise interaction scores and then merging them using the same thresholding scheme). That way, we can better understand the source of the strength of the method. - It would be interesting to see if Hessians still perform poorly even when explaining many-times differentiable networks, e.g. those with softplus rather than ReLU activations.

Correctness: The claims and methods largely seem sound. The empirical methodology is clever and seems correct, except for the lack of ArchDetect ablations mentioned above.

Clarity: On the positive side, many parts of the paper (especially the introduction) are well-written and compelling, and the visualizations are very clear. On the negative side, I found the definitions and explanations of the methods somewhat hard to digest. In order to understand each individual method, the reader is forced to parse through multiple sections and equations that are spaced apart (adding a lot of extraneous cognitive load). I think it would help a lot if each method could be extracted out into an algorithm box that fully shows all the steps in the computation.

Relation to Prior Work: Yes, clearly and extensively (especially with the history section in Appendix H).

Reproducibility: Yes

Additional Feedback: Just in terms of reproducibility, although I do think the work is reproducible, including code and a separate algorithm box would both help a lot. Overall, I feel this paper's strengths make it a clear accept and so I'm giving it a 7, but I do feel that for the camera-ready, it's critical to: - fix some of the clarity issues - add results for ArchAttribute with different methods for interaction detection - add more discussion about how to pick thresholds === Update in response to author feedback: I'm satisfied with the author feedback, especially the assumption 5 clarifications and the agreement to test out other interaction detectors with ArchAttribute. Looking forward to seeing this paper in its final form!

[Author Response · NeurIPS 2020]

Dear reviewers, thank you for your feedback. We'll take them all into account. Below are our responses to major points.

**Link between Interpretability and Ground Truth Annotations (from R5)** The link between interpretability and annotation labels comes from the concept of *coherence* for explanation evaluation [1]. Coherence means the consistency between an explanation and human prior belief. Here, the annotation label is the relevant prior belief since it is a human-defined and domain-specific label deemed most appropriate for a feature group. In fact, [2] demonstrated that attribution methods that correlate - or cohere - well with annotation labels also increased human trust in model predictions. We used this same annotation-based metric in our paper at lines 226-235. Practically-speaking, coherence is important so that users can agree with model interpretations. Without coherence, a user may inadvertently think that a model fails when it actually works, e.g. the motivational Fig. 1. We will include this discussion in our paper revision.

**Runtime Experiments (from R5)** We have runtime experiments in Appendix J (referenced on line 225). We will add a summary of the runtime result in the main paper. Our proposed method outperforms the state-of-the-art.

**Assumption 5 Clarifications**

- **Text Example (from R1):** The interactions {a, very, good} and {not, very} are captured by the contexts of $\mathbf{x}^\star$ and $\mathbf{x}'$. Assumption 5 argues that we can remove/omit any special higher-order (>3-way) interaction associated with both target and baseline tokens because such interactions are not very interpretable. It wouldn't make sense for a special 5-way interaction to be created by {_, a, very, good, _} or {not, _, very, _, _} when there could just be low-order interactions {a, very, good} and {not, very} created (which make more direct sense).
- **Synthetic Example (from R7):** This example creates an interaction between exactly 5 target values and 5 baseline values, so indeed Assumption 5 would be violated. The question worth considering here is: does it make sense for conditions to be placed on both target and baseline feature values? Perhaps an interaction between just 5 target values is already meaningful enough, which satisfies Assumption 5.
- **Testing Assumption 5 (from R7):** Our "Interaction Redundancy" experiment in Section 5.2 was designed to show how well Assumption 5 is satisfied via the redundancy of new interactions as the assumption is violated.

**On Faithfulness (from R4)** We did not claim that the ground truth metric is for faithfulness. In the interpretability literature, *faithfulness* and *coherence* are two different concepts. *Faithfulness* - as R4 suggested - refers to an accurate description of model decision-making [3]. For our discussion on *coherence*, please see our response on "Link between Interpretability and Ground Truth Annotations". Arguably, many interpretation methods are *faithful* according to their special properties, such as our method, Shapley Taylor Interaction Index, and Integrated Hessians via Axioms. Here, we are interested in annotation labels to evaluate *coherence*. (the word correlation metric was used out of respect of [2])

**Choice of Interaction Strength Threshold (from R7)** The threshold lies on a continuum between the interpretability and completeness tradeoff of an explanation - fundamental to Explainable AI [3]. A principled way to address this question is through an interactive visualization means, where a user is able to adjust the interaction strength threshold (with a slider UI) to understand its effect on explanations. If this is not an option, showing a small number of interactions (compared to the input size) is desirable for explanation simplicity [1]. An automatic way to determine the threshold is to use prediction performance gains of a surrogate interaction model as a guide, where this model is trained on the data samples from ArchDetect and interactions are added to the model until performance stops improving, similar to [4].

**Why these Axioms? (from R1)** Was the question about how axioms lead to better interpretability? The completeness axiom is designed to tell how much feature(s) impact predictions, and is widely desired for individual feature attribution and some feature interaction methods (Section 6). The Set Attribution axiom essentially allows us to obtain independent attribution scores $\phi(\mathcal{I}_i)$ for different disjoint feature sets $\mathcal{I}_i$ due to the additive structure of a function, so we can do the analysis of Fig. 2c on each of those feature sets, thereby leading to more interpretable results.

**Why subtract out the baseline in ArchAttribute? (from R2)** To satisfy axioms. The intuition for ArchAttribute's interpretability was shown in Fig. 2c. Indeed, our experiments in Table 2 showed that ArchAttribute is interpretable.

**How ArchDetect works for Image Classification? (from R2)** Please see discussions in Appendix B (and line 196).

**ArchAttribute with Other Interaction Detectors (from R7)** In interest of space, we will include these comparisons in paper revisions. One can still judge the different interaction detections in Appendix K by manually merging them.

**Hessians on many-times differentiable models (from R7)** Table 2 shows Integrated Hessians (IH) on BERT which uses many-times differentiable GELU activations. IH is computationally prohibitive to run for image classification.

[1]. Miller, T. *"Explanation in Artificial Intelligence: Insights from the Social Sciences"* in *Artificial Intelligence* **267** (Elsevier, 2019), 1–38. [2]. Jin, X. et al. *"Towards Hierarchical Importance Attribution: Explaining Compositional Semantics for Neural Sequence Models"* in *ICLR* (2020). [3]. Gilpin, L. H. et al. *"Explaining Explanations: An Overview of Interpretability of Machine Learning"* in *DSAA* (2018), 80–89. [4]. Tsang, M. et al. *"Feature Interaction Interpretability: A Case for Explaining Ad-Recommendation Systems via Neural Interaction Detection"* in *ICLR* (2020).


[Meta-Review · NeurIPS 2020]

The authors introduce Archipelago: a novel framework for detection and attribution of higher order feature interactions. Following the rebuttal, all reviewers recognized the strengths of the work. Please see their comments for details on ways to improve the paper, and incorporate the points from your rebuttal (note you have an extra page). Highlighting some of the aspects to be sure to include: The framing around interpretability Greater clarity, particularly on examples, methods and definitions Additional results Discussion on how to pick thresholds